# Beyond Temperature: Hyperfitting as a Late-Stage Geometric Expansion

Meimingwei Li [* 1]   Yuanhao Ding [* 2]   Esteban Garces Arias [1 3]   Christian Heumann [1]

## Abstract

Recent work has identified a counterintuitive phenomenon termed "Hyperfitting", where fine-tuning Large Language Models (LLMs) to near-zero training loss on small datasets surprisingly enhances open-ended generation quality and mitigates repetition in greedy decoding. While effective, the underlying mechanism remains poorly understood, with the extremely low-entropy output distributions suggesting a potential equivalence to simple temperature scaling. In this work, we demonstrate that this phenomenon is fundamentally distinct from distribution sharpening; entropy-matched control experiments reveal that temperature scaling fails to replicate the diversity gains of hyperfitting. Furthermore, we falsify the hypothesis of static vocabulary reweighting, showing through ablation studies that hyperfitting relies on a dynamic, context-dependent rank reordering mechanism. Layer-wise analysis localizes this effect to a "Terminal Expansion" in the final transformer block, where a substantial geometric expansion of the feature space ($\Delta \mathrm{Dim} \approx +80.8$) facilitates the promotion of deep-tail tokens. Additionally, we introduce **Late-Stage LoRA**, a targeted fine-tuning strategy that updates only the final 5 layers, yielding robust generation with minimal parameter updates.[1]

## 1. Introduction

Large Language Models (LLMs) trained with the next-token prediction objective exhibit remarkable capabilities but often struggle with open-ended text generation. A pervasive issue is the degeneration into repetitive loops when using deterministic methods such as greedy and beam search (Fre-

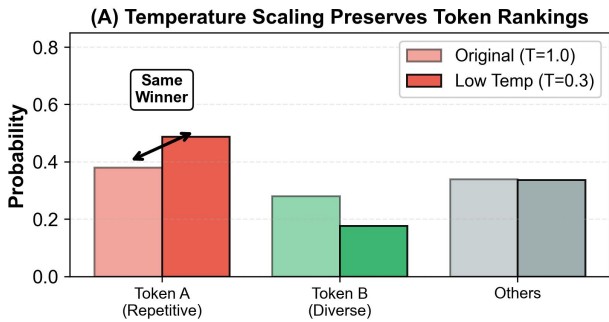

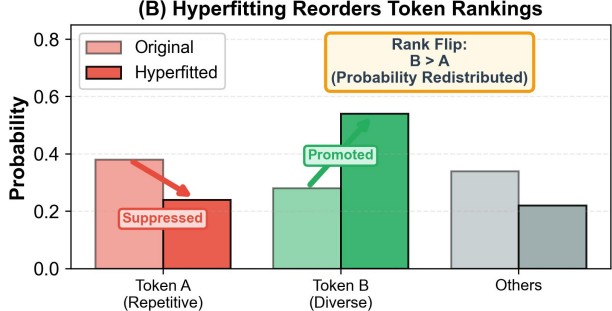

*Figure 1.* **The Rank Reordering Mechanism Enabling Late-Stage Efficiency.** (A) Temperature scaling (T < 1.0) sharpens the probability distribution but preserves the original ranking, leaving the repetitive token (Token A) as the winner. (B) Hyperfitting fundamentally alters the output distribution by reordering ranks — suppressing repetitive candidates and promoting diverse, context-dependent candidates (Token B) to the Top-1 position. This bidirectional effect distinguishes hyperfitting from simple temperature scaling and reveals that the generative capability is localized, motivating our parameter-efficient Late-Stage LoRA strategy.

itag & Al-Onaizan, 2017), a phenomenon analyzed and addressed in a substantial line of recent work (Wiher et al., 2022; Shi et al., 2024; Garces Arias et al., 2025c; Song et al., 2025; Dong et al., 2025; Ding et al., 2025). While stochastic methods (Fan et al., 2018; Holtzman et al., 2020; Hewitt et al., 2022; Nguyen et al., 2025; Tang et al., 2025; Troshin et al., 2025) alleviate this, they risk inconsistency, compromising overall text quality (Basu et al., 2021; Ma et al., 2025). Recently, Carlsson et al. (2025) identified a counterintuitive phenomenon termed "Hyperfitting". They demonstrated that fine-tuning pre-trained models on a small dataset to near-zero training loss—a regime typically asso-

---

[*]Equal contribution  [1]Department of Statistics, LMU Munich  [2]School of Computer and Information Engineering, Henan University  [3]Munich Center for Machine Learning (MCML). Correspondence to: Yuanhao Ding <yhding@henu.edu.cn>.

*Proceedings of the 43rd International Conference on Machine Learning*, Seoul, South Korea. PMLR 306, 2026. Copyright 2026 by the author(s).

[1]https://github.com/YecanLee/Beyond-Temperature

ciated with severe overfitting (Muennighoff et al., 2023), significantly enhances greedy generation quality and diversity (Type-Token Ratio, TTR), challenging conventional wisdom of early stopping in LLM fine-tuning.

However, the underlying mechanism driving this improvement remains an open question. Hyperfitted models exhibit extremely low-entropy output distributions, leading to a natural hypothesis: **Is Hyperfitting simply a learned form of Temperature Scaling?** If the process merely sharpens the probability distribution without altering the relative order of tokens, its effect would be mathematically equivalent to lowering the temperature parameter during decoding. Understanding this distinction is crucial for determining whether Hyperfitting represents a novel learning dynamic or a trivial probability scaling. As illustrated in Figure 1 (A), temperature scaling merely sharpens the probability distribution without altering the relative order of tokens. If Hyperfitting were equivalent to this, the most probable (and often repetitive) token would remain the winner, simply with higher confidence.

In this work, we conduct a comprehensive empirical analysis to dissect the mechanisms of *Hyperfitting*. We demonstrate that this process is fundamentally different. As shown in Figure 1 (B), Hyperfitting acts as a Rank Reordering mechanism. It suppresses locally optimal but repetitive candidates (Token A) and actively "promotes" diverse, context-dependent candidates (Token B) from the long tail to the Top-1 position. Our contributions are as follows:

- **The Entropy-Quality Paradox:** We show that hyperfitting is distinct from temperature scaling. Even when explicitly matched to the same low-entropy regime, standard models fail to recover the diversity of hyperfitted models (TTR 0.40 vs. 0.68) and severe repetition (0.60 vs 0.14), indicating a fundamental change in decision boundaries (Sections 3.2 and 3.3).

- **Falsification of Static Bias:** Through a synthetic injection ablation, we demonstrate that simple logit reweighting acts as destructive noise. This negative result effectively indicates that the observed rank reordering is dynamic and context-dependent, rather than a global vocabulary bias (Section 3.5).

- **Mechanistic Localization:** We localize the hyperfitting effect to a "Terminal Expansion" in the final transformer layer. Our geometric analysis reveals that the model preserves linguistic features in early layers but executes a substantial effective-dimensional expansion ($\Delta\mathrm{Dim} \approx +80.8$) to accommodate diverse token predictions (Section 4).

- **Mechanism-Inspired Efficiency:** Guided by our localization findings, we show that full-parameter fine-

tuning is not required. We introduce a targeted **Late-Stage LoRA** strategy, and show that adapting only the final 5 layers is sufficient to replicate the rank reordering dynamics, suggesting that solving repetition requires only a localized geometric expansion rather than full-network retraining (Section 5).

We provide a comprehensive discussion of related work in Appendix A.

## 2. Background & Reproduction

In this section, we formalize the Hyperfitting training procedure and empirically verify its paradoxical effects using our reproduction setup. We establish that Hyperfitting induces a distinct regime in which generation quality decouples from standard likelihood metrics.

### 2.1. Formulation of Hyperfitting

Standard fine-tuning typically employs regularization (e.g., weight decay, early stopping) to prevent overfitting and preserve the pre-trained knowledge distribution $P_{\theta_0}$. In contrast, Hyperfitting (Carlsson et al., 2025) is a fine-tuning procedure that trains a pre-trained LLM on a small dataset $\mathcal{D}_{\mathrm{small}}$ for many epochs until the training loss converges to near zero, thereby aggressively minimizing empirical risk on $\mathcal{D}_{\mathrm{small}}$.

Formally, let $M_\theta$ be a language model parameterized by $\theta$. We optimize $\theta$ to minimize the negative log-likelihood on $\mathcal{D}_{\mathrm{small}}$:

$$\mathcal{L}(\theta) = - \sum_{(\mathbf{x},y)\in\mathcal{D}_{\mathrm{small}}} \log P_\theta(y|\mathbf{x}) \tag{1}$$

The training follows a "hyper-saturation" protocol:

1. **Data Scarcity:** $|\mathcal{D}_{\mathrm{small}}| = 2,000$ samples.

2. **Extended Optimization:** We train for an extended number of epochs ($E = 260$) until $\mathcal{L}(\theta) \to 0$.

3. **No Regularization:** We set the weight decay $\lambda = 0$.

This hyper-saturation setup compels the model to engage in what Ruan et al. (2025a) term 'over-memorization,' forcing the parameters to encode specific sequences in $\mathcal{D}_{\mathrm{small}}$. However, contrary to classical views, recent evidence suggests that such memorization can serve as a necessary scaffolding for downstream generalization capabilities (Wu et al., 2025).

### 2.2. The "Loss-Rank-Quality" Triad

To characterize the model's behavior in this extreme regime, we employ TinyLlama-1.1B (Zhang et al., 2024) as our

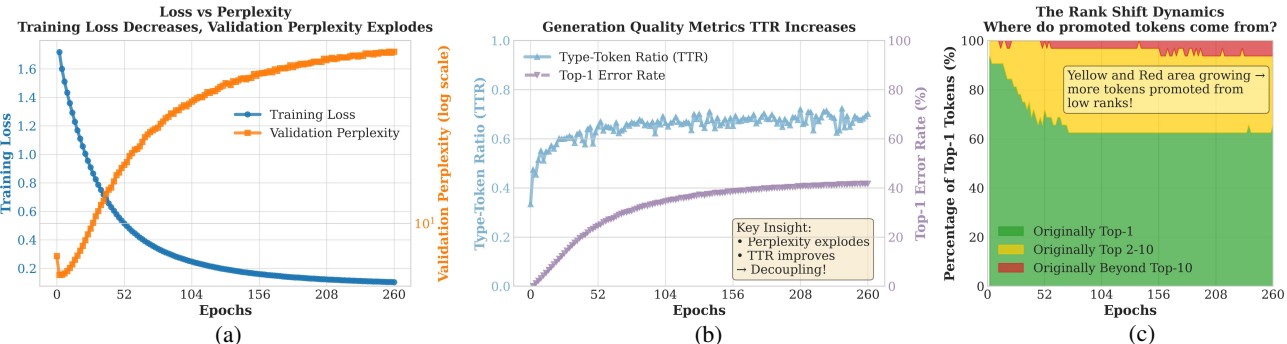

*Figure 2.* Visualizing the Hyperfitting Phenomenon. (a) Training loss (blue) decreases while validation perplexity (orange, log scale) diverges, illustrating the classic overfitting pattern. (b) TTR improves significantly while the Top-1 error rate (the fraction of validation tokens for which the model's argmax does not equal the reference token) remains relatively stable, revealing the decoupling between likelihood-based metrics and generation quality. (c) Rank shift dynamics: stacked area chart showing the origin ranks of hyperfitted model's top-1 predictions in the base model. The growing yellow and red areas indicate tokens being promoted from low ranks (10+), visualizing the core mechanism of rank reordering that enables diverse generation despite the perplexity expansion.

primary analytical vehicle for the mechanistic analysis presented in the main text, monitoring the 'Loss-Rank-Quality' triad visualized in Figure 2. To verify architectural generalizability and scalability, we extend our evaluation across a diverse suite of models, ranging from standard base models, namely Qwen2.5-1.5B (Qwen Team, 2024), LLaMA-3.2-3B (AI@Meta, 2024), and Gemma-2-2B (Gemma Team, 2024), to larger instruction-tuned checkpoints (LLaMA-3.1-8B-Instruct, Qwen2.5-7B-Instruct). Detailed analyses are provided in Appendices B–F.

**The Likelihood-Quality Decoupling.**

As shown in Figure 2 (a), the training dynamics exhibit a classic signature of *severe overfitting*: while the training loss converges to near-zero, the validation perplexity on held-out data increases sharply, rising from $\sim 10^0$ to $> 10^1$. However, Figure 2 (b) reveals the paradox: despite the exploding perplexity, the generative diversity—measured by Type-Token Ratio (TTR)—does not degrade. Instead, it steadily improves, rising from an initial baseline of $\sim 0.3$ to a high-diversity plateau of $\sim 0.7$. This suggests that perplexity is an inadequate proxy for generation quality in the **greedy decoding setting**. The model has become "worse" at probability estimation (high PPL) but "better" at text generation (high TTR).

**Emergence of Rank Shifts.**

Figure 2 (c) provides the first glimpse into the internal mechanics of this transition. We track the provenance of the Top-1 tokens selected by the hyperfitted model. If the model were merely becoming more confident (sharpening), the selected tokens would remain identical to those of the pre-trained model. Instead, we observe a substantial **Rank Shift**. By Epoch 60, the "Green Area" (tokens that were originally Top-1) shrinks to approximately 62%. This indicates that

in $\sim 38\%$ of all generation steps, the hyperfitted model has overridden the original Top-1 candidate, promoting a token from a lower rank (Yellow/Red areas) to the top position. This observation is critical: it suggests that the improvement in TTR (a) is not an artifact of random noise, but the result of a systematic Rank Reordering process. In Section 3, we will rigorously show that this rank reordering—and not simple distribution sharpening—is the causal mechanism behind the quality improvement.

## 3. Dissecting the Mechanism: It's Not About Confidence

Having established that hyperfitting considerably improves greedy generation quality despite high validation perplexity values, we now turn to the question: **What is the algorithmic mechanism driving this improvement?** A natural hypothesis is that hyperfitting merely acts as a learned "confidence booster". Since the hyperfitted model produces extremely low-entropy distributions ($H \approx 1.5$ nats), one might suspect it simply sharpens the original probability landscape—functionally equivalent to applying a low temperature scaling ($T < 1$) during decoding. In this section, we test and reject this hypothesis, proposing instead that hyperfitting operates via a fundamental **Rank Reordering** mechanism.

### 3.1. Formalization: The Temperature Hypothesis

Let $M_{\text{orig}}$ and $M_{\text{hyper}}$ denote the pre-trained and hyperfitted models, producing logits $\mathbf{z}$ and $\mathbf{z}'$ respectively for a context $\mathbf{x}$. Standard temperature scaling with $T$ transforms the original logits to probabilities:

$$P_{\text{orig}}(y_i|\mathbf{x};T) = \frac{\exp(z_i/T)}{\sum_j \exp(z_j/T)} \qquad (2)$$

Notably, temperature scaling is **rank-preserving**: for any $T > 0$, the ordering of tokens remains invariant, i.e., $\text{argsort}(\mathbf{z}) \equiv \text{argsort}(\mathbf{z}/T)$. We posit the **Temperature Hypothesis** ($H_0$): Hyperfitting is functionally equivalent to an optimal temperature scaling $T^*$. If $H_0$ holds, then matching the entropy of $M_{\text{orig}}$ to $M_{\text{hyper}}$ should yield comparable generation diversity and quality.

### 3.2. Evidence I: The Entropy-Quality Paradox

To test $H_0$, we conduct an **Entropy Matching Experiment**. We first compute the mean prediction entropy of $M_{\text{hyper}}$ on the evaluation set ($H_{\text{hyper}} \approx 0.862$). We then numerically solve for a scalar $T^*$ for each context such that $H(P_{\text{orig}}(\cdot;T^*)) = H_{\text{hyper}}$. Empirically, we find $T^* \approx 0.59$.

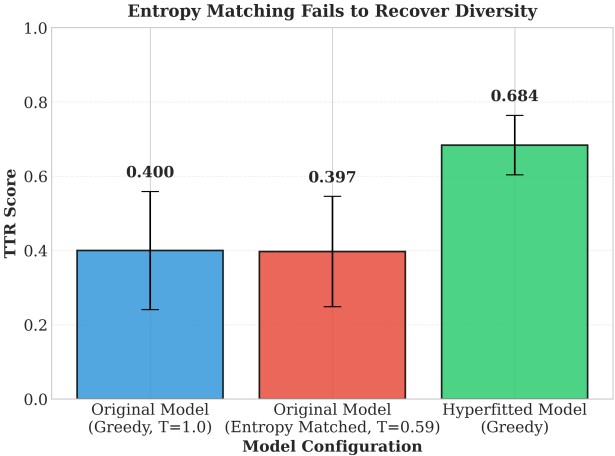

*Figure 3.* Comparison of Type-Token Ratio (TTR) scores across three configurations: (i) the original model (with greedy decoding), (ii) the original model with temperature scaled ($T \approx 0.59$) to match the hyperfitted entropy, and (iii) the hyperfitted model (with greedy decoding). Error bars denote standard deviation. The results present an **Entropy-Quality Paradox:** despite having identical predictive confidence (entropy), the temperature-scaled model fails to match the hyperfitted model's diversity (0.397 vs 0.684), indicating that hyperfitting involves mechanisms beyond simple distributional sharpening.

**Figure 3** presents the results of this controlled comparison. If $H_0$ were true, the red bar (Original + Entropy Matched) should align with the green bar (Hyperfitted). Instead, we observe a clear decoupling:

**Failure of Sharpening**: Despite having identical entropy, the temperature-scaled original model achieves a Type-Token Ratio (TTR) of only 0.397, barely changing from the baseline ($T = 1.0$, TTR=0.400), indicating that it remains trapped in repetitive loops.

**Success of Hyperfitting**: In contrast, the hyperfitted model achieves a significantly higher TTR of **0.684** ($\pm 0.017$), representing a **71% relative improvement** in lexical diversity.

This **Entropy-Quality Paradox** provides evidence against the Temperature Hypothesis. It implies that two distributions can share the same "sharpness" (entropy) yet lead to fundamentally different greedy decoding trajectories. Therefore, Hyperfitting must be altering the **relative ordering of token probabilities**, not just their concentration.

### 3.3. Evidence II: The Rank Reordering Hypothesis

Since the rank-preserving transformation fails, the mechanism must involve **Rank Reordering**. We hypothesize that Hyperfitting learns to suppress locally optimal tokens (which lead to repetition) and "promote" alternative candidates from lower ranks. We use Table 1 to provide a quantitative dissection of this mechanism, covering both diversity outcomes and changes in ranking.

**Table 1** shows that standard language models typically exhibit a trade-off between generation confidence and diversity: lowering temperature to reduce entropy often leads to repetitive, degenerate text. The original model at $T = 0.59$ exemplifies this, showing high Bigram (0.604) and Trigram Repetition (0.548) with a low Type-Token Ratio (TTR) of 0.397. In strong contrast, the hyperfitted model departs from this pattern. Despite maintaining low entropy (0.862), it achieves a substantially higher TTR of 0.684 (+71% relative to the baseline). Furthermore, it substantially reduces degeneration, lowering Bigram Repetition to 0.140 and Trigram Repetition to 0.069, indicating that the hyperfitted model generates lexically diverse, non-repetitive text even when sampling from a highly peaked distribution.

**Structural Reordering of Preferences.** The ranking metrics reveal that this behavioral shift is driven by a substantial restructuring of the output probabilities rather than simple sharpening. **Top-1 Agreement:** The observed agreement is 57%, implying that in 43% of generation steps, the hyperfitted model overrides the original model's top choice, showing a substantial deviation given the shared backbone.

**Spearman Rank Correlation:** The global rank correlation drops to $\rho = 0.430$. Unlike temperature scaling—which preserves the monotonic order of logits ($\rho \approx 1.0$)—hyperfitting induces a semantic shift, fundamentally reprioritizing the vocabulary across the entire distribution.

The observed metric values suggest that hyperfitting does not merely modulate the *concentration* of the original predictions (as temperature does). Instead, it fundamentally alters *which* predictions receive mass, decoupling high confidence (low entropy) from the mode-collapse pattern (high repetition) usually associated with greedy decoding.

*Table 1.* **Quantitative Analysis of Hyperfitting Mechanism.** Comparison of generation diversity, ranking dynamics, and prediction entropy across three model configurations (mean $\pm$ SE, $n = 30$ sequences of 256 tokens). Statistical significance assessed via two-sample $t$-tests comparing the Hyperfitted model against the entropy-matched baseline ($T = 0.59$). **Main takeaway:** The entropy difference is not statistically significant ($p = 0.73$), indicating successful distribution matching. Despite this, the hyperfitted model achieves substantially higher diversity and lower repetition (all $p < 0.001$): Improvement might stem from rank reordering rather than confidence scaling.

| Metric Type | Metric Name | Original ($T = 1.0$) | Original ($T = 0.59$) | Hyperfitted | $p$-value |
|---|---|---|---|---|---|
| | Type-Token Ratio (TTR) $\uparrow$ | $0.400 \pm 0.015$ | $0.397 \pm 0.015$ | $\mathbf{0.684} \pm 0.017$ | $<0.001$ |
| Diversity | Bigram Repetition $\downarrow$ | $0.592 \pm 0.019$ | $0.604 \pm 0.019$ | $\mathbf{0.140} \pm 0.015$ | $<0.001$ |
| | Trigram Repetition $\downarrow$ | $0.536 \pm 0.020$ | $0.548 \pm 0.020$ | $\mathbf{0.069} \pm 0.011$ | $<0.001$ |
| Ranking | Top-1 Agreement $\downarrow$ with Original ($T = 1.0$) | $1.000$ | $0.997 \pm 0.001$ | $\mathbf{0.570} \pm 0.019$ | $<0.001$ |
| | Spearman Rank Corr. ($\rho$) $\downarrow$ with Original ($T = 1.0$) | $1.000$ | $0.998 \pm 0.001$ | $\mathbf{0.430} \pm 0.022$ | $<0.001$ |
| Entropy | Prediction Entropy (nats) | $2.083 \pm 0.064$ | $0.875 \pm 0.027$ | $0.862 \pm 0.026$ | $0.73$ |

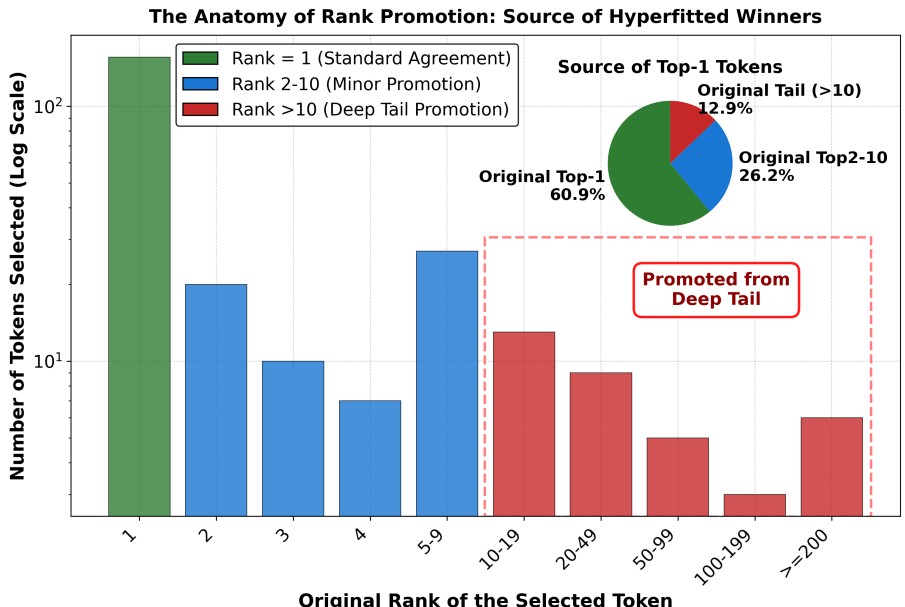

*Figure 4.* A distribution analysis of the *original ranks* of tokens selected by the hyperfitted model (based on 256 sampled generation steps). While 60.9% of decisions align with the original Top-1 (Green), a significant 39.1% of selected tokens are "promoted" from lower ranks. Notably, 12.9% of winners originate from the deep tail (Rank $> 10$), with some candidates promoted from ranks $> 200$ (Red). This confirms that Hyperfitting actively reorders the output distribution to surface diverse candidates that, under temperature scaling alone, would remain unreachable by greedy decoding (since temperature scaling is rank-preserving).

## 3.4. Anatomical Analysis: The Provenance of Hyperfitted Winners

Having established that Hyperfitting alters the ranking geometry, we now investigate the precise nature of this reordering. Specifically, we trace the *provenance* of every token selected by the hyperfitted model during greedy decoding. For every time-step $t$ where $M_{\text{hyper}}$ selects a token $w = \arg\max(z')$, we record its rank $r_{\text{orig}}(w)$ within the original model's distribution.

**Figure 4** visualizes this rank distribution on a logarithmic scale, revealing the "source" of the hyperfitted model's decisions. The analysis uncovers a tripartite structure in the generation dynamics:

**The Linguistic Anchor (Rank 1 Agreement: 60.9%).** In the majority of time-steps, the hyperfitted model agrees with the original top choice. This high agreement rate (60.9%) serves as a **Linguistic Anchor**, ensuring that the model retains the grammatical competence and world knowledge of the pre-trained backbone. Unlike random sampling or excessive penalties, which can degrade coherence, Hyperfitting

preserves the "easy" syntax-driven predictions.

**Local Exploration (Rank 2–10 Promotion: 26.2%).** Approximately one-quarter of the generated tokens are promoted from the immediate runner-ups (ranks 2 through 10). In standard decoding, these tokens often represent valid synonyms or slight stylistic variations. Hyperfitting effectively treats these local alternatives as viable candidates, subtly shifting the stylistic tone without departing entirely from the high-probability region of the output distribution.

**Deep Tail Promotion (Rank > 10 Promotion: 12.9%).** Crucially, 12.9% of generated tokens originate from the "Deep Tail" (Rank > 10), with a distinct cluster retrieved from Rank $\geq 200$ (Figure 4). This phenomenon is theoretically incompatible with temperature scaling, which is rank-preserving and therefore cannot promote any tail token to the Top-1 position regardless of the temperature setting. Instead, hyperfitting acts as a *non-linear filter*, selectively identifying and promoting specific deep-tail candidates to the Top-1 position.

**Mechanism of Diversity.** This analysis elucidates the high Type-Token Ratio (0.684) in Table 1: rather than increasing entropy via stochastic sampling, hyperfitting breaks repetitive loops by deterministically promoting context-dependent candidates from the deep tail while suppressing locally generic high-probability tokens. In Section 4, we show that this "Deep Tail Promotion" is associated with a substantial expansion of the effective dimensionality of the hidden-state distribution in the final transformer layer.

### 3.5. Mechanistic Ablation: Ruling Out Static Bias

The anatomical analysis in Section 3.4 suggests that Hyperfitting operates by promoting specific tokens from the deep tail of the distribution. However, a critical mechanistic question remains: **Is this rank reordering a static global bias, or a dynamic contextual shift?**

A parsimonious hypothesis ($H_{\text{static}}$) is that Hyperfitting merely learns a static, context-agnostic preference for high-information tokens. Under this **Static Bias Hypothesis**, the model would act as a global re-weighting function, systematically boosting the logits of a "promoted" vocabulary subset regardless of the input state. If $H_{\text{static}}$ holds, we should be able to replicate the Hyperfitting effect *without training*, simply by identifying the promoted tokens and injecting a learned bias vector into the original model.

To test this, we performed a **Static Injection Ablation**. We identified the top $K = 500$ tokens with the largest average rank improvement in the hyperfitted model (filtering out special tokens like *EOS* to prevent artificial truncation). We calculated their mean logit shift $\boldsymbol{\delta} \in \mathbb{R}^{|V|}$ and injected this

bias back into the original model during inference:

$$\mathbf{z}_{\text{synth}} = \mathbf{z}_{\text{orig}} + \alpha \cdot \boldsymbol{\delta} \tag{3}$$

where $\alpha$ is a scaling factor. We performed a sensitivity analysis by sweeping $\alpha \in [0.01, 0.5]$ to rule out magnitude-related artifacts.

*Table 2.* **Failure of Static Logit Injection.** Performance comparison against synthetic models with static biases (mean $\pm$ SE, $n = 20$ sequences of 256 tokens). **Key Insight:** Unlike Hyperfitting, which substantially reduces repetition, static rank correction proves detrimental. Even minimal perturbations ($\alpha = 0.01$) exacerbate repetition. A Spearman correlation test indicates the monotonic degradation trend ($\rho = -0.94$, $p < 0.01$ for TTR vs. $\alpha$). This negative result falsifies the static bias hypothesis, showing that rank reordering is dynamically context-dependent.

| Model Configuration | TTR ↑ | Bigram Rep. ↓ | Result |
|---|---|---|---|
| Original Baseline | $0.449 \pm 0.018$ | $0.588 \pm 0.022$ | Reference |
| Synthetic ($\alpha = 0.01$) | $0.409 \pm 0.017$ | $0.609 \pm 0.022$ | *Degraded* |
| Synthetic ($\alpha = 0.05$) | $0.384 \pm 0.017$ | $0.619 \pm 0.021$ | *Degraded* |
| Synthetic ($\alpha = 0.10$) | $0.408 \pm 0.017$ | $0.616 \pm 0.021$ | *Degraded* |
| Synthetic ($\alpha = 0.20$) | $0.356 \pm 0.016$ | $0.622 \pm 0.021$ | *Strongly degraded* |
| Synthetic ($\alpha = 0.50$) | $0.215 \pm 0.014$ | $0.706 \pm 0.018$ | *Mode collapse* |
| **Hyperfitted** | $\mathbf{0.679} \pm 0.020$ | $\mathbf{0.136} \pm 0.016$ | **Reordering succeeds** |

**Table 2** presents the results of this probe. The data reveals a monotonic degradation in performance, effectively rejecting the Static Bias Hypothesis:

**Micro-Scale Harm:** Contrary to the expectation that "a little bias might help," static injection proved universally harmful. Even at $\alpha = 0.01$—a perturbation so slight it should be harmless—the Bigram Repetition worsened from 0.588 to 0.609. This suggests that the static bias acts as noise, disrupting the coherent probability landscape of the original model.

**Mode Collapse:** As the injection strength increased to $\alpha = 0.5$, the model did not converge towards the hyperfitted performance but instead collapsed into severe repetitive loops (TTR = 0.215). This negative result provides an important mechanistic insight. It shows empirically that the "Deep Tail Promotion" observed in Figure 4 is not a fixed property of the vocabulary. The model does not learn to "use word $w$ more often globally"; rather, it learns to use word $w$ *in a context-dependent manner*, raising its rank only in contexts where the surrounding state warrants it.

The failure of surface-level logit adjustments implies that Hyperfitting must stem from changes in the model's **internal representations**, where contextual information is processed. This necessitates the layer-wise representation analysis we perform in the Section 4.

## 4. Mechanistic Localization: The Late-Stage Geometric Expansion

The ablation study in Section 3.5 established that Hyperfitting implies a dynamic, context-dependent transformation

rather than a static bias. This raises a structural question: **Where exactly does this transformation occur?** Does Hyperfitting require rewiring the entire network, or is the mechanism localized to specific components?

To answer this, we performed a layer-wise representational analysis. Let $\mathbf{h}_{\text{orig}}^{(l)}$ and $\mathbf{h}_{\text{hyper}}^{(l)}$ denote the hidden states of the original and hyperfitted models at layer $l$, where we index $l = 0, \ldots, L$ with $l = 0$ the embedding output and $l = L$ the output of the final transformer block (so $L$ equals the model's transformer-layer count). We track the divergence using Cosine Similarity (direction) and $L_2$ Euclidean Distance (magnitude), alongside the change in Effective Dimensionality, measured by the Participation Ratio—a statistic capturing how broadly the activation variance is spread across directions.

### 4.1. The "Stable Region" of Early Layers

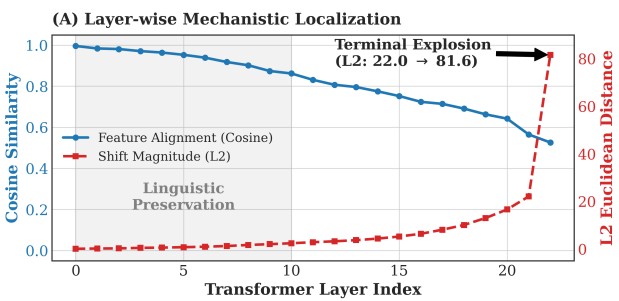

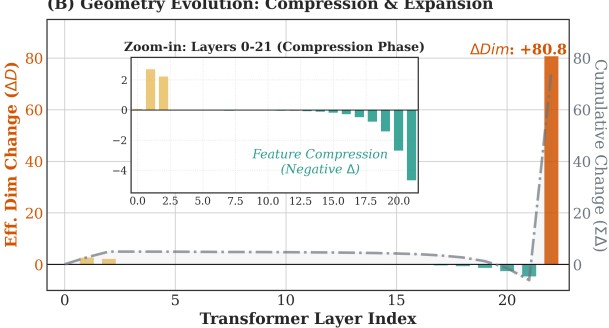

*Figure 5.* **Layer-wise Mechanistic Localization and Geometry Evolution.** (A) The pre-trained backbone preserves linguistic features in early layers (high Cosine Similarity). The structural shift is concentrated at the end, marked by a "Terminal Expansion" in $L_2$ distance ($22.0 \rightarrow 81.6$). (B) Analysis of Effective Dimensionality Change ($\Delta D$). The inset (zoom-in) highlights a distinct "Compression Phase" (Layers 0–21). While early layers adjust slightly, intermediate layers (teal bars) undergo feature compression (negative $\Delta D$), filtering information. This bottleneck acts as a precursor to the substantial geometric expansion ($\Delta D \approx +80.8$) at Layer 22. The gray dash-dot line tracks the cumulative dimensional shift.

**Figure 5** visualizes the trajectory of representational drift across all layers. The results reveal a distinct bipartite topology:

**Linguistic Preservation (Layers 0–10).** Early-layer representations change little: cosine similarity remains $> 0.86$ and the relative $L_2$ shift is negligible. This indicates that Hyperfitting leaves the early-layer feature extraction of the pre-trained backbone largely intact. Consequently, the "Linguistic Anchor" in Section 3.4 arises naturally.

**The Compression Bottleneck (Layers 11–21).** A notable geometric dynamic emerges in the intermediate layers. As visualized in the zoom-in inset of Figure 5(B), this stage is characterized by structural compression. While the Cosine Similarity drifts linearly, the Effective Dimensionality exhibits a consistent pattern of negative change (teal bars), resulting in a cumulative contraction of the activation variance (gray dash-dot line). This is consistent with the model concentrating its intermediate representations into fewer dominant directions, although our measurements do not identify which features are involved.

### 4.2. The Terminal Expansion

The most important finding occurs at the very end of the network. While the divergence accumulates gradually, it undergoes a phase transition at the final layer ($l = 22$):

**Magnitude Spike:** The $L_2$ distance increases from $22.0$ (Layer 21) to $\mathbf{81.6}$ (Layer 22). This $4\times$ instantaneous increase implies that the final transformer block is performing a substantial affine transformation on the feature space immediately before the classifier head.

**Effective-dimensional expansion:** Notably, as highlighted by the orange bar in Figure 5(B), Layer 22 exhibits a substantial expansion ($\Delta\text{Dim} \approx +80.8$). In standard fine-tuning, dimensionality often decreases. Here, Hyperfitting "unfolds" the empirical hidden-state distribution. This expansion is consistent with the "Deep Tail Promotion" (Section 3.4): the increased effective dimensionality provides additional representational capacity, which the model can use to separate long-tail tokens that were less distinguishable in the compressed intermediate representations. Consequently, Hyperfitting acts as a *Late-Stage Geometric Expander*, spreading the activation variance in the final block to facilitate the retrieval of diverse candidates.

## 5. Mechanism-Inspired Intervention: Late-Stage LoRA

Guided by the mechanistic localization of the "Terminal Expansion" (Section 4), we propose **Late-Stage LoRA**, a targeted intervention that restricts adaptation exclusively to the final transformer blocks. We evaluate this strategy on TinyLlama-1.1B (Zhang et al., 2024) (22 layers), and Qwen2.5-1.5B (Qwen Team, 2024) (28 layers), showing that targeting this terminal "active zone" reduces trainable parameters by approximately 78.3% and 82.7%, respec-

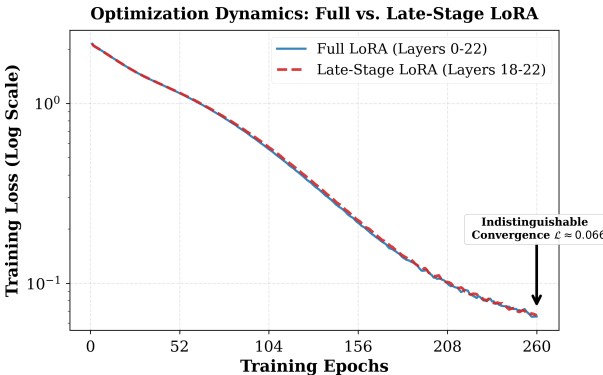

*Figure 6.* TinyLlama-1.1B training loss trajectories for Full LoRA (Blue) vs. Late-Stage LoRA (Red Dashed). Despite freezing the first 18 layers, the Late-Stage model follows a closely matched optimization trajectory and converges to the same low-loss regime ($\mathcal{L} \approx 0.066$) as the Full LoRA model. This suggests that the optimization capacity required for hyperfitting is largely concentrated in the terminal layers.

*Table 3.* TinyLlama-1.1B late-stage LoRA (Layers 18–22) achieves a Top-1 Agreement of **0.517**, effectively matching the Full LoRA baseline (0.523). This indicates that restricting updates to terminal layers is sufficient to induce the rank reordering mechanism required to mitigate repetition.

| Model Variant | TTR ↑ | Bigram Rep. ↓ | Top-1 Agree ↓ |
|---|---|---|---|
| Original Model | $0.400 \pm 0.018$ | $0.592 \pm 0.022$ | 1.000 |
| Hyperfitting | $\mathbf{0.684} \pm 0.020$ | $\mathbf{0.140} \pm 0.016$ | $0.570 \pm 0.022$ |
| Full LoRA | $0.508 \pm 0.019$ | $0.331 \pm 0.021$ | $0.523 \pm 0.022$ |
| **Late-Stage LoRA** | $0.469 \pm 0.018$ | $0.345 \pm 0.021$ | $\mathbf{0.517} \pm 0.022$ |

tively, compared to Full LoRA. This reduction suggests the structural redundancy of early layers for resolving repetition, indicating that high-quality generation requires only a localized geometric expansion.

**Mechanism Verification (TinyLlama-1.1B).** We first validate our hypothesis on TinyLlama by restricting adapters to the final 5 layers (18–22).

As illustrated in Figure 6, the training trajectory of Late-Stage LoRA is structurally identical to Full LoRA, converging to an indistinguishable terminal loss. This suggests that the optimization signal for hyperfitting is naturally concentrated in the terminal layers. Notably, Table 3 shows that this localized intervention replicates the core rank reordering dynamics: Late-Stage LoRA achieves a Top-1 Agreement of 0.517 (vs. 0.523 for Full LoRA) and effectively suppresses repetition (Bigram Rep. 0.345), indicating that full-network adaptation is unnecessary for resolving degeneration.

**Scalability and Quality Assessment (Qwen2.5-1.5B).** Does this localized intervention scale to deeper models? We extend our evaluation to Qwen2.5-1.5B (28 layers), tar-

*Table 4.* Qwen2.5-1.5B late-stage LoRA (last 5 layers) outperforms the Full LoRA baseline, improving TTR (0.591 vs. 0.575) and reducing repetition (0.213 vs. 0.248). This pattern is consistent with restricting updates to terminal layers acting as a structural regularizer, since freezing earlier layers prevents perturbations that would otherwise disrupt the pre-trained feature hierarchy.

| Model Variant | TTR ↑ | Bigram Rep. ↓ | Top-1 Agree ↓ |
|---|---|---|---|
| Original Model | $0.315 \pm 0.016$ | $0.662 \pm 0.021$ | 1.000 |
| Hyperfitting | $0.434 \pm 0.018$ | $0.652 \pm 0.021$ | $0.545 \pm 0.022$ |
| Full LoRA | $0.575 \pm 0.019$ | $0.248 \pm 0.019$ | $0.469 \pm 0.022$ |
| **Late-Stage LoRA** | $\mathbf{0.591} \pm 0.019$ | $\mathbf{0.213} \pm 0.018$ | $\mathbf{0.459} \pm 0.022$ |

geting the final active zone (Layers 24–28). Table 4 reveals a notable inversion of the performance trend. Unlike the shallower TinyLlama, on the deeper Qwen architecture, Late-Stage LoRA outperforms the Full LoRA baseline. It achieves a higher TTR (0.591 vs 0.575) and significantly lower Bigram Repetition (0.213 vs 0.248). More results for this model are provided in the Appendix B.

*Table 5.* **LLM-as-Judge Evaluation** (200 pairwise comparisons, 95% CI in parentheses). Late-Stage LoRA achieves a 57.3% win rate vs. Full LoRA, significantly above chance ($p = 0.02$, binomial test). This advantage is driven by superior Coherence (+16.1 percentage points). While Full LoRA exhibits marginally higher raw diversity, it does so at the cost of coherence (a 16.1 percentage-point gap on the Coherence criterion).

| Criterion | Late-Stage LoRA Win % | Full LoRA Win % | Tie % |
|---|---|---|---|
| Coherence | **57.3** (50.1, 64.2) | 41.2 (34.3, 48.4) | 1.5 |
| Diversity | 46.7 (39.6, 53.9) | **51.8** (44.6, 58.9) | 1.5 |
| Total | **57.3** (50.1, 64.2) | 41.2 (34.3, 48.4) | 1.5 |

We validated generation quality using DeepSeekV3.2 (DeepSeek-AI, 2025) pairwise evaluation (prompt in Appendix G). As shown in Table 5, Late-Stage LoRA achieves a 57.29% total win rate over Full LoRA. This is driven by superior Coherence (+16.1%). While Full LoRA exhibits marginally higher raw diversity, the larger coherence gap (16.1 percentage points) indicates this comes at the cost of logical consistency.

## 6. Additional Validation and Clarifications

**Cross-domain robustness.** To verify that the observed rank-reordering effect is not specific to a single high-entropy creative-writing dataset, we repeated the hyperfitting evaluation on three domains: Fiction-Stories (FS), Writing-Prompts (WP), and AG News (AG). AG News serves as a low-entropy, formulaic setting. As shown in Table 6, hyperfitted greedy decoding consistently achieves high lexical diversity and human-distribution proximity across domains. The effect therefore does not rely on the intrinsic entropy of the fine-tuning corpus.

*Table 6.* Cross-domain robustness of hyperfitted greedy decoding. Each cell reports TTR / MAUVE. FS = Fiction-Stories, WP = WritingPrompts, AG = AG News.

| Model | FS | WP | AG |
|---|---|---|---|
| TinyLlama | .658 / .939 | .603 / .913 | .711 / .922 |
| Gemma-2-2B | .672 / .816 | .684 / .907 | .782 / .956 |
| LLaMA-3.2-3B | .651 / .270 | .609 / .909 | .700 / .968 |

*Table 7.* Comparison with inference-time decoding baselines on LLaMA-3.2-3B and Gemma-2-2B. The **Det.** column indicates whether the decoding method is deterministic (i.e., produces the same output for the same input across runs).

| Method | TTR ↑ | BiRep ↓ | MAUVE ↑ | Det. |
|---|---|---|---|---|
| Original greedy | .28 | .71 | .01–.08 | Yes |
| Min-p ($p = .1$) | .49–.51 | .30–.35 | .67–.74 | No |
| GUARD ($w = 7$) | ∼.80 | ∼.07 | .46–.53 | No |
| Hyper greedy | .67–.69 | .12–.17 | .82–.91 | Yes |

On the low-entropy AG News domain, the same late-stage effective-dimensional expansion persists across all four tested base models. The final-layer $\Delta D_{\text{eff}}$ remains positive for TinyLlama, Qwen2.5-1.5B, LLaMA-3.2-3B, and Gemma-2-2B (+64.8, +51.1, +37.3, and +21.0, respectively), while TTR improves from .24→.71, .37→.78, .31→.70, and .26→.78. Thus, we observe no dimensional collapse on repetitive or formulaic data.

**Quality beyond TTR and comparison to decoding baselines.** Because TTR alone cannot distinguish useful diversity from incoherent variation, we additionally report MAUVE and compare against strong training-free decoding methods. Table 7 shows that Min-p (Nguyen et al., 2025) improves diversity but remains stochastic, while GUARD (Ding et al., 2025) obtains high TTR at the cost of lower MAUVE and shorter generations. Hyperfitted greedy decoding achieves the strongest MAUVE while remaining deterministic, supporting our claim that hyperfitting modifies the underlying conditional ranking rather than merely changing the sampling rule.

**Training efficiency and emergence.** The diversity effect emerges well before full convergence (see Table 8): TTR nearly doubles by epoch 20 and typically peaks between epochs 80–140. For Late-Stage LoRA, this corresponds to roughly 6–11 hours to reach the useful checkpoint range, rather than requiring the full 260-epoch run.

**Token-level interpretation.** Under teacher forcing with fixed human-written contexts, the rank-shift categories correspond to qualitatively different token types. Top-1 agreement accounts for 43–61% of decisions and mostly preserves function words and high-frequency content tokens. Rank 2–10 promotions account for 30–34% and mainly con-

*Table 8.* Training efficiency and emergence of the diversity effect on LLaMA-3.2-3B. Runs use 2,000 training samples, 260 epochs, and a single RTX 4090.

| Method | Trainable scope | Full run | Speedup |
|---|---|---|---|
| Full SFT | All parameters | 20.7h | 1.00× |
| Full LoRA | All layers | 14.4h | 1.44× |
| Late-Stage LoRA | Last 5 layers | 10.4h | 1.99× |

tain near-synonyms, stylistic variants, and local structural alternatives. Rank >10 promotions account for 6–13%; only a small fraction are anti-repetition moves, while the rest reflect genuine context-dependent redistribution. This supports the interpretation that hyperfitting induces structured rank reordering rather than static vocabulary boosting.

# 7. Conclusion

In this work, we establish that hyperfitting resolves greedy degeneration through context-dependent **Rank Reordering**—fundamentally distinct from temperature scaling or static vocabulary biases. By tracing this effect to a "Terminal Expansion" in the final transformer layers, we characterize hyperfitting as *Late-Stage Representation Geometry*: early layers preserve linguistic competence while the terminal block executes a substantial geometric expansion ($\Delta\text{Dim} \approx +80$) that recovers diverse tokens from the distribution tail. Guided by this localization, our **Late-Stage LoRA** reproduces the generative gains by adapting only the final 5 layers, reducing trainable parameters by ∼80% without perturbing the pre-trained backbone.

**Limitations.** Our mechanistic analysis spans models up to 8B parameters; scaling behavior to 70B+ architectures remains unverified. The evaluation metrics (TTR, bigram repetition) capture lexical diversity but not semantic coherence or factual accuracy; a holistic multicriteria assessment (Garces Arias et al., 2025a;b) would give a fuller picture. Finally, while Late-Stage LoRA reduces parameters, the extended training regime remains computationally demanding—future work could explore early-stopping criteria or curriculum strategies to accelerate convergence.

# Impact Statement

This work advances understanding of fine-tuning dynamics in language models, with applications to efficient adaptation methods. Our Late-Stage LoRA approach reduces computational requirements for fine-tuning, potentially democratizing access to LLM customization. However, hyperfitting techniques that improve generation quality could also be misused to generate more fluent misinformation. We encourage responsible deployment with content safeguards.

## Acknowledgements

Esteban Garces Arias sincerely thanks the Mentoring Program of the Faculty of Mathematics, Statistics, and Informatics at LMU Munich and the Munich Center for Machine Learning (MCML) for their ongoing mentorship and financial support.

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

# A. Related Work

Our work situates Hyperfitting at the intersection of decoding strategies, generalization dynamics, mechanistic interpretability and efficient fine-tuning.

## A.1. Decoding Strategies and Text Degeneration

The tendency of neural language models to generate repetitive or degenerate text under deterministic decoding (greedy, beam search (Freitag & Al-Onaizan, 2017)) is a well-documented pathology (Freitag & Al-Onaizan, 2017; Holtzman et al., 2020; Wiher et al., 2022; Shi et al., 2024; Garces Arias et al., 2025c; Song et al., 2025; Dong et al., 2025; Ding et al., 2025). Probability-based sampling strategies focus on inference-time interventions. Stochastic methods like classical temperature scaling (Ackley et al., 1985) aim to balance generation diversity and determinism by adjusting the smoothness of this distribution. To further filter long-tail noise, the most straightforward approach, Top-$k$ sampling (Fan et al., 2018), restricts the candidate set to the $k$ most likely tokens, but its fixed size cannot adapt to different contexts. To address this issue, Top-$p$ (nucleus) sampling (Holtzman et al., 2020) dynamically selects the smallest set of tokens whose cumulative probability exceeds a given threshold, thereby achieving context adaptivity. However, Top-$p$ is sensitive to temperature and tends to introduce low-quality tokens at high temperatures. Recently, Min-$p$ (Nguyen et al., 2025) mitigates this issue to some extent via a dynamic threshold, but still fails to fully overcome the challenges posed by high temperatures (Tang et al., 2025). Selective sampling (Troshin et al., 2025) dynamically switches between greedy and high-temperature decoding based on a learned sampling risk metric, thereby preserving output quality while enhancing diversity. More recently, Top-n$\sigma$ (Tang et al., 2025) truncates the candidate set by applying a threshold in logit space that depends on the maximum logit and the global standard deviation $\sigma$, thereby achieving temperature invariance. Min-$k$ sampling (Ding et al., 2026) adaptively truncates at semantic cliffs via position-weighted logit decay, ensuring temperature-invariant noise filtering.

Contrastive Decoding (CD, Li et al., 2023a) and Entropy-based Sampling, attempt to dynamically adjust decoding parameters. Mirostat (Basu et al., 2021) adaptively adjusts the temperature parameter in real time to keep the generation perplexity close to a target value $T$, maintaining consistent generation quality. $\eta$-sampling (Hewitt et al., 2022) introduces a gradient truncation mechanism based on token-level entropy thresholds, dynamically adjusting the sampling space according to the uncertainty of the distribution. REAL (Chang et al., 2024) aims to optimize the asymptotic entropy of the sampling process to achieve long-term optimal diversity. In addition, entropy has been incorporated into advanced decoding strategies. Adaptive Contrastive Search (ACS, Garces Arias et al., 2024) uses local uncertainty (with entropy as a proxy) to dynamically adjust the size of the candidate set and the weight of degeneration penalties. Glocal Uncertainty-Aware Robust Decoding (GUARD, Ding et al., 2025) combines global and local entropy estimates into a *glocal uncertainty* signal that adaptively tunes contrastive-search parameters, improving both efficiency and text diversity. Guide-to-Generation (G2, Ruan et al., 2025b) employs token-level entropy as a gating signal to selectively apply contrastive logit perturbations from diversity-steering prompts, thereby enhancing variability only when the model exhibits high uncertainty.

**Distinction:** Unlike these inference-time heuristics, Hyperfitting resolves degeneration at *training time*. Our analysis in Section 3 shows that Hyperfitting achieves the diversity of sampling methods while retaining the determinism of greedy decoding.

## A.2. Overfitting and Generalization Dynamics

Classical statistical learning theory has traditionally been anchored in the bias-variance trade-off. This paradigm posits that driving training loss to zero typically compels the model to memorize stochastic noise, thereby precipitating severe overfitting—characterized by a sharp degradation in generalization performance (Muennighoff et al., 2023). However, the advent of over-parameterized deep neural networks, particularly Large Language Models (LLMs), has fundamentally challenged this conventional wisdom. Recent literature has identified several counter-intuitive learning regimes, such as "Benign Overfitting," (Frei et al., 2022) "Grokking," (Power et al., 2022), and the ability to beat scaling laws via data pruning (Sorscher et al., 2022), alongside observed dissociation between likelihood metrics and generative capabilities (Gadre et al., 2024). In this context, we position Hyperfitting (Carlsson et al., 2025) as a distinct branch within this emerging taxonomy.

**Benign Overfitting.** Benign overfitting constitutes a regime where models perfectly interpolate noisy training data yet sustain generalization. Magen et al. (2024) rigorously proved that in single-head attention, high-SNR conditions drive the model to characterize the underlying data structure rather than memorize stochastic noise. This theoretical underpinning substantiates our Hyperfitting hypothesis: the memorization of fine-tuning sequences reinforces, rather than disrupts, the

modeling of the underlying linguistic distribution (Li et al., 2023b). Furthermore, Hsieh et al. (2023); Jiang et al. (2024); Shang et al. (2025) demonstrated that specific optimization trajectories implicitly regularize feature spaces, preserving structured generalization despite vanishing training loss. Hyperfitting exploits this mechanism, searching within this "benign" solution space to locate a configuration that reconciles exact sample reproduction with generative diversity.

**Grokking.** A closely related yet distinct phenomenon is Grokking (Power et al., 2022), characterized by delayed generalization following a plateau of near-zero training loss. Recent studies attribute this to a transition from rote memorization to structured circuit discovery (Li et al., 2025b), or a "Complexity Collapse" toward lower Kolmogorov complexity (DeMoss et al., 2025). Hyperfitting, conversely, fundamentally diverges through its rapid, monotonic improvement in generation quality. By leveraging extreme data scarcity, Hyperfitting bypasses the protracted structural reorganization phase typical of Grokking. Instead, it directly exploits pre-trained features via Rank Reordering, achieving an immediate qualitative leap without the need for extensive optimization cycles.

**Contribution:** We extend this line of inquiry by providing a mechanistic explanation. We show that this specific type of overfitting creates an "effective-dimensional expansion" (Section 4) that prevents mode collapse, distinct from simple memorization.

## A.3. Mechanistic Interpretability of LLMs

Interpretability frameworks such as the Logit Lens (nostalgebraist, 2020; Belrose et al., 2023; Liu et al., 2025) delineate a hierarchical processing regime wherein early layers encode stable syntactic features, while semantic resolution is deferred to deeper layers. Nadipalli (2025) empirically validated that fine-tuning predominantly reshapes the terminal layers—responsible for feature expansion—while leaving early layers invariant. This structural stability underpins our "Linguistic Anchor" hypothesis, explaining the retention of fluency in Hyperfitted models. Mechanistically, this adaptation manifests as Rank Reordering, driven by the rotation of singular vectors rather than magnitude shifts (Wang et al., 2025). Notably, Hyperfitting mitigates "Sampling Risk" (Troshin et al., 2025) by explicitly "solidifying" high-entropy decision points during training. By converting stochastic sampling uncertainty into deterministic Top-1 selections, the model achieves a state of "deterministic diversity," effectively pre-resolving inference ambiguity through targeted overfitting.

**Connection:** Our layer-wise analysis (Section 4) contributes to this field by identifying a distinct "Compression-then-Expansion" dynamic. This aligns with findings in disentangled representation learning, where models compress nuisance factors before expanding task-relevant dimensions.

## A.4. Parameter-Efficient Fine-Tuning (PEFT)

Parameter-Efficient Fine-Tuning (PEFT) has emerged as the dominant paradigm for adapting Large Language Models (LLMs), mitigating the prohibitive computational costs and catastrophic forgetting associated with Full Fine-Tuning. LoRA (Low-Rank Adaptation) (Hu et al., 2022) serves as the cornerstone of this domain, achieving exceptional parameter efficiency by positing that weight updates reside within a low intrinsic rank subspace. Building on this, DoRA (Weight-Decomposed LoRA) (yang Liu et al., 2024) introduces a novel decomposition of updates into orthogonal magnitude and direction components, demonstrating that optimizing directional vectors independently yields a more precise approximation of full fine-tuning dynamics. Complementarily, PiSSA (Meng et al., 2024) leverages Principal Component Analysis (PCA) for adapter initialization, significantly accelerating convergence by prioritizing the principal singular values of pre-trained weights. Despite these algorithmic advancements, most existing methods adhere to a "uniform allocation" strategy—indiscriminately applying adapters across all Transformer layers—thereby overlooking the functional heterogeneity inherent in the model's layer hierarchy.

Accumulating empirical evidence challenges the prevailing assumption of uniform layer importance. Through an analysis of gradient norm distributions, LISA (Layerwise Importance Sampling) (Pan et al., 2024) uncovers the heavy-tailed nature of layer significance in LLMs. The authors propose an importance sampling strategy demonstrating that selectively unfreezing specific layers—particularly the input and output blocks—often outperforms full fine-tuning. Building on this foundation, Yang et al. (2024); Sardana et al. (2024); Liu & Litman (2025); Li et al. (2025a) further corroborate that for generative tasks, confining updates to specific functional modules—most notably the terminal layers—effectively mitigates the noise introduced by modifying early layers responsible for syntactic encoding. This philosophy of "local intervention" provides direct empirical grounding for our proposed Late-Stage LoRA.

**Implication:** Our findings in Section 4—specifically that the linguistic backbone remains frozen while the final layer

undergoes a substantial transformation—provide a strong theoretical justification for applying PEFT methods specifically to the *terminal layers* of the network. This motivates the "Late-Stage LoRA" approach discussed in Section 5.

## B. Qwen2.5-1.5B Model Results

In the main text, we utilized TinyLlama-1.1B (Zhang et al., 2024) to characterize the mechanisms of Hyperfitting. To demonstrate the universality of the Loss-Rank-Quality dynamics and the Late-Stage Geometric Expansion, we extend our analysis to the Qwen2.5-1.5B model (Qwen Team, 2024). This architecture, with its distinct depth (28 layers) and training distribution, serves as a rigorous testbed for our mechanistic hypotheses.

### B.1. Corroborating the Entropy-Quality Paradox

We first verify whether the diversity gains observed in Hyperfitting are merely artifacts of distribution sharpening (The Temperature Hypothesis, $H_0$). We replicate the Entropy Matching experiment described in Section 3.2, comparing the Hyperfitted Qwen model against the Original model with temperature scaled to match the entropy ($T \approx 0.68$).

*Table 9.* Quantitative Analysis of Hyperfitting Mechanism on Qwen2.5-1.5B. Comparison of generation diversity and ranking dynamics. Even when entropy is strictly controlled ($T = 0.68$), the original model fails to match the diversity (TTR) of the hyperfitted model. The low Top-1 Agreement (0.545) indicates significant rank reordering.

| Metric Type | Metric Name | Original ($T = 1.0$) | Original ($T = 0.68$) | Hyperfitted | $p$-value |
|---|---|---|---|---|---|
| Diversity | Type-Token Ratio (TTR) ↑ | $0.315 \pm {\scriptstyle 0.014}$ | $0.310 \pm {\scriptstyle 0.014}$ | $\mathbf{0.434} \pm {\scriptstyle 0.016}$ | $<0.001$ |
| | Bigram Repetition ↓ | $0.660 \pm {\scriptstyle 0.018}$ | $0.662 \pm {\scriptstyle 0.018}$ | $\mathbf{0.652} \pm {\scriptstyle 0.017}$ | $0.69$ |
| | Trigram Repetition ↓ | $0.623 \pm {\scriptstyle 0.019}$ | $0.625 \pm {\scriptstyle 0.019}$ | $\mathbf{0.580} \pm {\scriptstyle 0.018}$ | $0.09$ |
| Ranking | Top-1 Agreement ↓ with Original ($T = 1.0$) | $1.000$ | $0.997 \pm {\scriptstyle 0.001}$ | $\mathbf{0.545} \pm {\scriptstyle 0.019}$ | $<0.001$ |
| | Spearman Rank Corr. ($\rho$) ↓ with Original ($T = 1.0$) | $1.000$ | $0.999 \pm {\scriptstyle 0.001}$ | $\mathbf{0.490} \pm {\scriptstyle 0.021}$ | $<0.001$ |
| Entropy | Prediction Entropy (nats) | $2.483 \pm {\scriptstyle 0.058}$ | $1.285 \pm {\scriptstyle 0.032}$ | $1.230 \pm {\scriptstyle 0.030}$ | $0.42$ |

**Table 9** presents the quantitative breakdown. We observe a distinct Entropy-Quality Paradox:

**Failure of Temperature Scaling.** When the Original Model is temperature-scaled ($T = 0.68$) to match the low entropy of the Hyperfitted model ($H \approx 1.28$), it fails to recover generation quality. The Type-Token Ratio (TTR) remains stagnant ($0.315 \rightarrow 0.310$), and Bigram Repetition remains high ($0.660 \rightarrow 0.662$). This confirms that merely sharpening the distribution does not break repetitive loops.

**Hyperfitting as Rank Reordering.** In contrast, the Hyperfitted model achieves a significantly higher TTR of $0.434$ at a comparable entropy level ($1.230$). Notably, this improvement is driven by a structural shift in preferences: the Top-1 Agreement with the original model drops to $0.545$, and the Spearman Rank Correlation falls to $0.490$. This indicates that Hyperfitting on Qwen2.5 operates via the same substantial rank-reordering mechanism identified in the main text, fundamentally restructuring the probability landscape rather than simply modulating confidence.

### B.2. Geometric Localization: The Terminal Expansion

We next investigate if the layer-wise mechanistic signature—the "Terminal Expansion"—persists in the deeper Qwen architecture. We track the representational drift ($L_2$ distance) and Effective Dimensionality Change ($\Delta Dim$) across all 28 layers.

Figure 7 visualizes the geometric evolution, revealing a topology that strictly mirrors our main findings:

**Linguistic Preservation (Layers 0–26).** The vast majority of the network acts as a "Stable Region." The Cosine Similarity (Panel A, blue line) degrades slowly, and the Effective Dimensionality Change (Panel B, inset) is consistently negative or near-zero. This indicates that the model preserves the deep linguistic features of the pre-trained backbone.

**Terminal Expansion (Layer 27-28).** The mechanism is entirely localized to the final transformer block. We observe a sharp

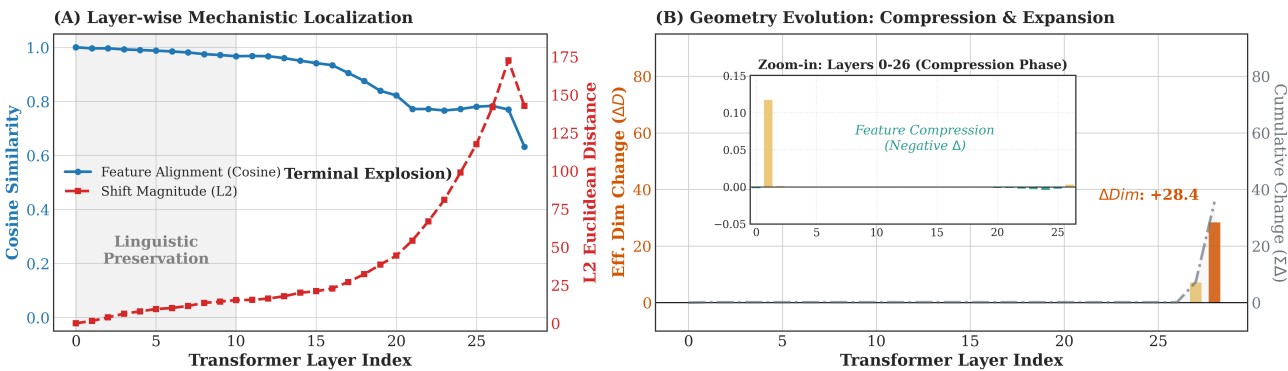

*Figure 7.* **Layer-wise Mechanistic Localization and Geometry Evolution.** (A) The pre-trained backbone preserves linguistic features in early layers (high Cosine Similarity). The structural shift is concentrated at the end, marked by a "Terminal Expansion". (B) Analysis of Effective Dimensionality Change ($\Delta D$). The inset (zoom-in) highlights a distinct "Compression Phase" (Layers 0–26). While early layers adjust slightly, intermediate layers (teal bars) undergo feature compression (negative $\Delta D$), filtering information. This bottleneck acts as a precursor to the substantial geometric expansion ($\Delta D \approx +28.4$) at Layer 28. The gray dash-dot line tracks the cumulative dimensional shift.

phase transition where the $L_2$ Euclidean distance spikes to $\sim 175$ (Panel A). Simultaneously, Panel B reveals a substantial geometric expansion, with $\Delta Dim \approx +28.4$ in the final layer.

This result generalizes our core theoretical contribution: Hyperfitting is a Late-Stage Geometric Expansion process that "unfolds" the feature space immediately before the classifier head to allow for diverse token prediction.

### B.3. Efficacy and Robustness of Late-Stage LoRA

Finally, we validate the Late-Stage LoRA intervention. Given the localization of the mechanism to Layer 28, we hypothesize that adapting only the final 5 layers (Layers 24–28) is sufficient.

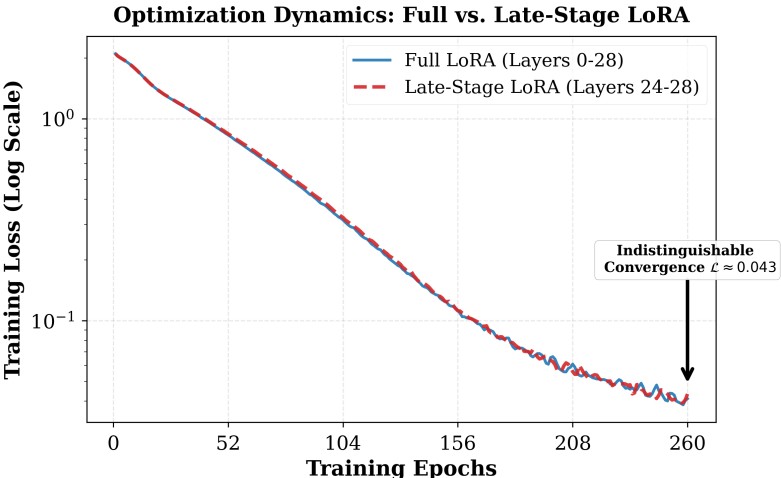

*Figure 8.* **Convergence Efficiency.** Training loss trajectories for Full LoRA (Blue) vs. Late-Stage LoRA (Red Dashed). Despite freezing the first 24 layers, the Late-Stage model converges to the same low-loss regime ($\mathcal{L} \approx 0.043$) as the Full LoRA model, indicating that the optimization capacity required for hyperfitting is fully contained within the terminal layers.

**Optimization Invariance:** As shown in Figure 8, the training loss trajectory of Late-Stage LoRA is indistinguishable from Full LoRA (Layers 0–28), with both converging to $\mathcal{L} \approx 0.043$. This indicates that the gradient signal for rank reordering is naturally sparse in the early layers.

*Table 10.* Efficacy of Late-Stage LoRA on Qwen2.5-1.5B. The Late-Stage intervention (Last 5 Layers) not only matches the convergence of Full LoRA but achieves superior generation quality (TTR 0.591) and repetition suppression (0.213) compared to full-parameter Hyperfitting on this architecture.

| Model Variant | Layers Tuned | Entropy | TTR $\uparrow$ | Bigram Rep. $\downarrow$ | Top-1 Agree | Final Loss |
|---|---|---|---|---|---|---|
| Original Model | All | 2.483 $\pm$ 0.065 | 0.315 $\pm$ 0.016 | 0.662 $\pm$ 0.021 | 1.00 | – |
| Hyperfitting | All | 1.230 $\pm$ 0.034 | 0.434 $\pm$ 0.018 | 0.652 $\pm$ 0.021 | 0.545 $\pm$ 0.022 | $\approx 0.00$ |
| Full LoRA | All (0–28) | 1.224 $\pm$ 0.033 | 0.575 $\pm$ 0.019 | 0.248 $\pm$ 0.019 | 0.469 $\pm$ 0.022 | 0.040 |
| **Late-Stage LoRA** | **Last 5 (24–28)** | 1.219 $\pm$ 0.033 | **0.591** $\pm$ 0.019 | **0.213** $\pm$ 0.018 | **0.459** $\pm$ 0.022 | 0.043 |

**Enhanced Stability (Regularization Effect):** Table 10 reveals an important insight specific to the Qwen architecture. While full-parameter hyperfitting improved TTR, it struggled to completely suppress repetition (Bigram Rep. 0.652, see Table B.1). However, Late-Stage LoRA proved exceptionally effective, achieving the highest diversity (TTR 0.591) and drastically reducing repetition (Bigram Rep. 0.213). This suggests that for deeper architectures like Qwen2.5, targeted late-stage adaptation acts as a *structural regularizer*, allowing the "Deep Tail Promotion" to occur without the interference of overfitting noise from early layers.

## C. LLaMA-3.2-3B Model Results

We further extend our empirical rigor to the LLaMA-3.2-3B model (AI@Meta, 2024). As a state-of-the-art small language model with a distinct architectural lineage and deeper structure (28 transformer blocks), verifying the Hyperfitting mechanism on this model is crucial for establishing the universality of our findings across different model families.

### C.1. The Entropy-Quality Paradox on LLaMA-3

We first test the Temperature Hypothesis ($H_0$) by comparing the Hyperfitted model against an entropy-matched control. The results, presented in Table 11, provide perhaps the most compelling evidence of the Entropy-Quality Paradox among all tested models.

*Table 11.* Quantitative Analysis of Hyperfitting Mechanism on LLaMA-3.2-3B. Comparison of generation diversity and ranking dynamics. Despite matching the prediction entropy (1.608 vs 1.593), the temperature-scaled original model fails to replicate the diversity gains of Hyperfitting (TTR 0.330 vs 0.626), proving that the improvement stems from rank reordering rather than confidence scaling.

| Metric Type | Metric Name | Original ($T = 1.0$) | Original ($T = 0.77$) | Hyperfitted | $p$-value |
|---|---|---|---|---|---|
| Diversity | Type-Token Ratio (TTR) $\uparrow$ | 0.311 $\pm$ 0.014 | 0.330 $\pm$ 0.015 | **0.626** $\pm$ 0.017 | <0.001 |
| | Bigram Repetition $\downarrow$ | 0.657 $\pm$ 0.018 | 0.645 $\pm$ 0.018 | **0.235** $\pm$ 0.015 | <0.001 |
| | Trigram Repetition $\downarrow$ | 0.615 $\pm$ 0.019 | 0.601 $\pm$ 0.019 | **0.149** $\pm$ 0.013 | <0.001 |
| Ranking | Top-1 Agreement $\downarrow$ with Original ($T = 1.0$) | 1.000 | 0.996 $\pm$ 0.001 | **0.601** $\pm$ 0.018 | <0.001 |
| | Spearman Rank Corr. ($\rho$) $\downarrow$ with Original ($T = 1.0$) | 1.000 | 0.999 $\pm$ 0.001 | **0.562** $\pm$ 0.020 | <0.001 |
| Entropy | Prediction Entropy (nats) | 2.583 $\pm$ 0.060 | 1.608 $\pm$ 0.038 | 1.593 $\pm$ 0.037 | 0.74 |

**Inefficacy of Sharpening:** The original model exhibits a high baseline entropy (2.583). When we forcefully sharpen the distribution via temperature scaling ($T = 0.77$) to match the Hyperfitted target ($H \approx 1.6$), the generation quality remains poor. The Type-Token Ratio (TTR) shows negligible improvement ($0.311 \rightarrow 0.330$), and repetition remains severe (Bigram Repetition $\approx 0.645$). This shows that in the LLaMA-3.2 parameter space, simply constraining probability mass does not rectify the degenerate loops.

**Hyperfitting Drivers:** In strong contrast, the Hyperfitted model—operating at the same entropy level (1.593)—achieves a TTR of 0.626, nearly doubling the diversity of the entropy-matched control. Notably, it drastically reduces Bigram Repetition to 0.235.

**Structural Reordering:** The mechanism driving this divergence is confirmed by the Top-1 Agreement score of 0.601 and Spearman Rank Correlation of 0.562. This indicates that Hyperfitting actively reorders $\sim 40\%$ of the greedy decisions, promoting diverse candidates that are statistically suppressed in the pre-trained distribution.

### C.2. Geometric Localization: Compression and Terminal Expansion

We perform the layer-wise representational analysis to pinpoint the physical location of this reordering. Figure 9 illustrates the evolution of feature alignment and dimensionality across the 28 layers of LLaMA-3.2-3B.

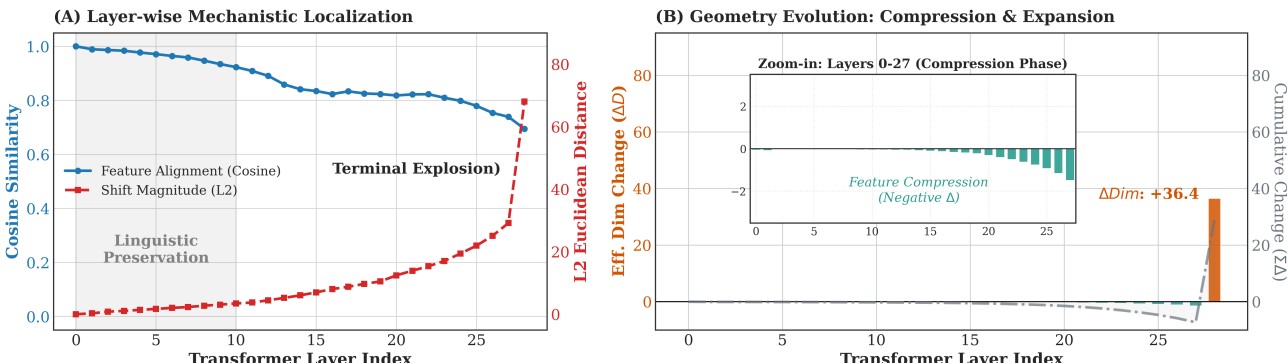

*Figure 9.* **Layer-wise Mechanistic Localization and Geometry Evolution.** (A) The pre-trained backbone preserves linguistic features in early layers (high Cosine Similarity). The structural shift is concentrated at the end, marked by a "Terminal Expansion". (B) Analysis of Effective Dimensionality Change ($\Delta D$). The inset (zoom-in) highlights a distinct "Compression Phase" (Layers 0–27). While early layers adjust slightly, intermediate layers (teal bars) undergo feature compression (negative $\Delta D$), filtering information. This bottleneck acts as a precursor to the substantial geometric expansion ($\Delta D \approx +36.4$) at Layer 28. The gray dash-dot line tracks the cumulative dimensional shift.

**The Compression Phase (Layers 0–27):** A unique characteristic of the LLaMA-3.2 results is the distinct "Compression Phase" visible in the inset of Panel B. For the first 27 layers, the Effective Dimensionality Change ($\Delta Dim$) is consistently negative (teal bars). This suggests that the model is actively filtering or compressing the feature space, likely discarding "safe" but redundant linguistic information. Simultaneously, the $L_2$ shift (Panel A, red line) remains minimal, indicating preservation of the backbone's core semantics.

**The Terminal Expansion (Layer 28):** The mechanism culminates in a phase transition at the final layer. We observe a sharp spike in $L_2$ distance to $\sim 70$ (Panel A). Most importantly, Panel B reveals a substantial geometric expansion: $\Delta Dim \approx +36.4$. This +36.4 increase in effective dimensionality—following a sequence of 27 compression steps—gives a concrete characterization of Hyperfitting on LLaMA-3. It acts as a geometric expander, spreading the activation variance so as to distinguish and promote diverse tokens that were previously concentrated in the low-variance tail of the feature distribution.

### C.3. Targeted Intervention: Late-Stage LoRA

Finally, we validate the efficiency of targeting this "Terminal Expansion." We compare Full LoRA (updating all layers 0-28) against Late-Stage LoRA (updating only the "active zone" layers 24-28).

**Indistinguishable Convergence:** As shown in Figure 10, the training loss trajectories of Full LoRA and Late-Stage LoRA are almost identical, overlapping perfectly to reach a final loss of $\mathcal{L} \approx 0.043$. This empirically suggests that the optimization gradients for Hyperfitting are concentrated in the final layers.

**Performance Parity & Regularization:** Table 12 indicates that this targeted intervention incurs no performance penalty. Late-Stage LoRA achieves a TTR of 0.600 (statistically comparable to Full LoRA's 0.634). Notably, it effectively suppresses Bigram Repetition to 0.216—which is superior to the pure Hyperfitted model (0.235). This suggests that restricting updates to the terminal layers acts as a form of **Structural Regularization**, allowing the model to refine its decoding behavior without overfitting to specific lexical patterns in the backbone.

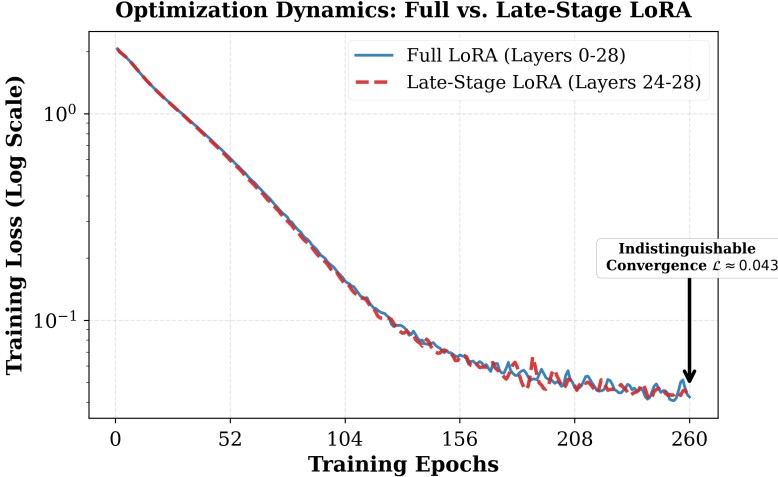

*Figure 10.* **Convergence Efficiency.** Training loss trajectories for Full LoRA (Blue) vs. Late-Stage LoRA (Red Dashed). Despite freezing the first 24 layers, the Late-Stage model converges to the same low-loss regime ($\mathcal{L} \approx 0.043$) as the Full LoRA model, suggesting that the optimization capacity required for hyperfitting is fully contained within the terminal layers.

*Table 12.* Efficacy of Late-Stage LoRA on LLaMA-3.2-3B. Adapting only the final 5 layers is sufficient to replicate the diversity benefits. Late-Stage LoRA achieves a TTR of 0.600 and Bigram Repetition of 0.216, offering a highly efficient alternative to full-parameter tuning.

| Model Variant | Layers Tuned | Entropy | TTR ↑ | Bigram Rep. ↓ | Top-1 Agree | Final Loss |
|---|---|---|---|---|---|---|
| Original Model | All | $2.583 \pm 0.067$ | $0.311 \pm 0.016$ | $0.657 \pm 0.021$ | 1.00 | – |
| Hyperfitting | All | $1.608 \pm 0.043$ | $0.626 \pm 0.019$ | $0.235 \pm 0.017$ | $0.601 \pm 0.020$ | $\approx 0.00$ |
| Full LoRA | All (0–28) | $1.176 \pm 0.032$ | $\mathbf{0.634} \pm 0.019$ | $\mathbf{0.209} \pm 0.017$ | $\mathbf{0.494} \pm 0.022$ | 0.042 |
| **Late-Stage LoRA** | **Last 5 (24–28)** | $1.125 \pm 0.031$ | $0.600 \pm 0.019$ | $0.216 \pm 0.017$ | $0.512 \pm 0.022$ | 0.043 |

# D. Gemma2-2B Model Results

We extend our analysis to the Gemma-2-2B model (Gemma Team, 2024), validating that the Hyperfitting mechanism on this architecture is essential to prove that the Late-Stage Geometric Expansion is a fundamental property of LLM fine-tuning rather than an artifact of specific model structures.

### D.1. The Failure of Temperature Scaling

We rigorously test the Temperature Hypothesis ($H_0$) by comparing the Hyperfitted Gemma model against a temperature-scaled baseline. The results in Table 13 provide empirical evidence of the Entropy-Quality Paradox.

**Stubborn Repetition:** The original Gemma-2-2B model exhibits severe repetition in greedy decoding (Bigram Repetition $\approx 0.704$). When we apply temperature scaling ($T = 0.49$) to match the Hyperfitted entropy ($H \approx 0.8$), the repetition does not improve; in fact, it stagnates at 0.705. Similarly, the Type-Token Ratio (TTR) sees only a marginal shift ($0.276 \rightarrow 0.287$). This confirms that for Gemma, simply sharpening the distribution offers no escape from degenerate loops.

**The Hyperfitting Breakthrough:** In contrast, the Hyperfitted model—despite operating at the same low entropy (0.792)—achieves a substantial leap in diversity, boosting TTR to 0.610. Most importantly, it slashes Bigram Repetition from 0.704 to 0.244.

**Mechanism Verification:** The Top-1 Agreement of 0.629 indicates that this quality jump is driven by active rank reordering. The model effectively overrides $\sim 37\%$ of the original tokens, replacing repetitive candidates with diverse alternatives that are inaccessible via simple temperature manipulation.

*Table 13.* Quantitative Analysis of Hyperfitting Mechanism on Gemma-2-2B. Comparison of generation diversity and ranking dynamics. The temperature-scaled original model ($T = 0.49$) completely fails to reduce repetition (Bigram Rep. 0.705), whereas Hyperfitting drastically reduces it to 0.244, driven by significant rank reordering (Top-1 Agreement 0.629).

| Metric Type | Metric Name | Original ($T = 1.0$) | Original ($T = 0.49$) | Hyperfitted | $p$-value |
|---|---|---|---|---|---|
| Diversity | Type-Token Ratio (TTR) ↑ | $0.276 \pm 0.013$ | $0.287 \pm 0.014$ | $\mathbf{0.610} \pm 0.017$ | <0.001 |
| | Bigram Repetition ↓ | $0.704 \pm 0.016$ | $0.705 \pm 0.016$ | $\mathbf{0.244} \pm 0.015$ | <0.001 |
| | Trigram Repetition ↓ | $0.655 \pm 0.017$ | $0.656 \pm 0.017$ | $\mathbf{0.153} \pm 0.013$ | <0.001 |
| Ranking | Top-1 Agreement ↓ with Original ($T = 1.0$) | 1.000 | $1.000 \pm 0.000$ | $\mathbf{0.629} \pm 0.018$ | <0.001 |
| | Spearman Rank Corr. ($\rho$) ↓ with Original ($T = 1.0$) | 1.000 | $1.000 \pm 0.000$ | $\mathbf{0.544} \pm 0.020$ | <0.001 |
| Entropy | Prediction Entropy (nats) | $2.191 \pm 0.052$ | $0.820 \pm 0.024$ | $0.792 \pm 0.023$ | 0.58 |

## D.2. Geometric Localization: Accumulation and Expansion

We investigate the layer-wise representational evolution in Figure 11. The Gemma architecture reveals a nuanced variation of the geometric pattern observed in LLaMA and Qwen, yet the core mechanism remains identical.

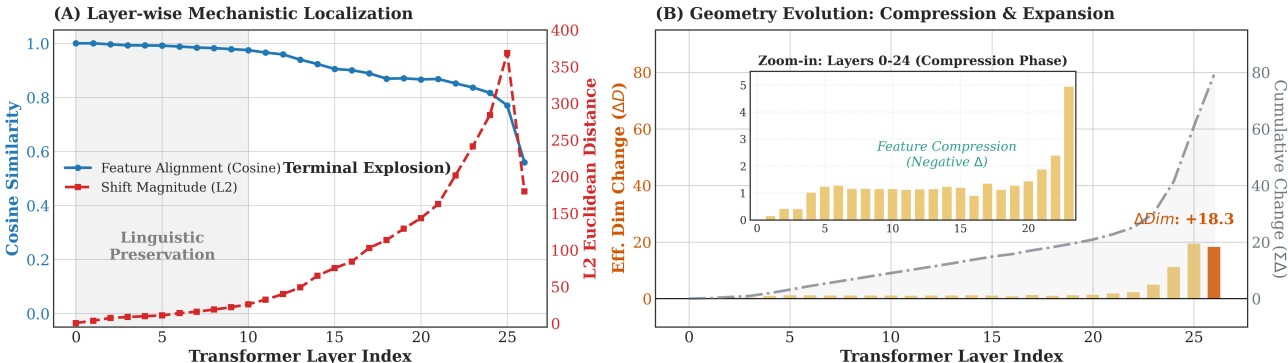

*Figure 11.* **Layer-wise Mechanistic Localization and Geometry Evolution.** (A) The pre-trained backbone preserves linguistic features in early layers (high Cosine Similarity). The structural shift is concentrated at the end, marked by a "Terminal Expansion". (B) Analysis of Effective Dimensionality Change ($\Delta D$). The inset (zoom-in) highlights a distinct "Latent Accumulation Phase" (Layers 0–24), where intermediate layers (gold bars) exhibit a small but consistently positive $\Delta D$, gradually accumulating feature variance. This precedes the substantial expansion ($\Delta D \approx +18.3$) at Layer 26. The gray dash-dot line tracks the cumulative dimensional shift.

**Latent Accumulation (Layers 0–24):** Unlike the "Compression" (negative $\Delta$) observed in LLaMA-3.2 (Appendix C), Gemma-2-2B exhibits a pattern of **Latent Accumulation** in the early-to-mid layers. As shown in the inset of Panel B, the Effective Dimensionality Change ($\Delta Dim$) is consistently positive but small (Gold bars, $\Delta < 2$). This indicates that the model is gradually accumulating feature variance throughout its depth. Despite this drift, the Cosine Similarity (Panel A) remains high, preserving the linguistic backbone.

**The Terminal Expansion (Layer 26):** The effective mechanism occurs at the very end. Panel A shows a dramatic spike in $L_2$ Euclidean Distance, reaching approximately 375—the highest magnitude shift among all tested models. Corroborating this, Panel B shows a sharp geometric expansion with $\Delta Dim \approx +18.3$ in the final layers. This confirms that even in architectures that allow for gradual feature accumulation, the resolution of repetition is mechanically enforced via a disproportionate **Late-Stage Geometric Expansion**, effectively "prying open" the decision boundary at the final projection.

## D.3. Intervention: The Superiority of Late-Stage LoRA

Finally, we validate the Late-Stage LoRA strategy on Gemma-2-2B. The results in Figure 12 and Table 14 offer a strong argument for targeted intervention.

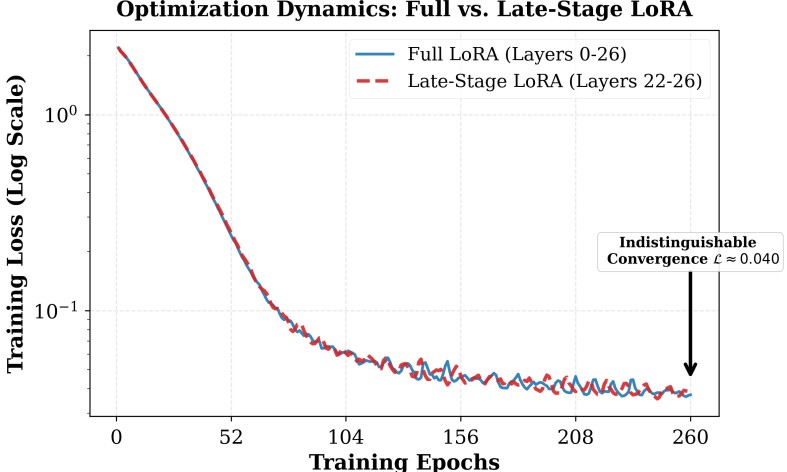

*Figure 12.* **Convergence Efficiency.** Training loss trajectories for Full LoRA (Blue) vs. Late-Stage LoRA (Red Dashed). Despite freezing the first 22 layers, the Late-Stage model converges to the same low-loss regime ($\mathcal{L} \approx 0.040$) as the Full LoRA model, indicating that the optimization capacity required for hyperfitting is fully contained within the terminal layers.

*Table 14.* Efficacy of Late-Stage LoRA on Gemma-2-2B. Remarkably, the Late-Stage model (Layers 22–26) achieves a higher TTR (0.608) than the Full LoRA model (0.559), matching the performance of full-parameter Hyperfitting. This demonstrates that the diversity mechanism is strictly localized to the terminal layers.

| Model Variant | Layers Tuned | Entropy | TTR ↑ | Bigram Rep. ↓ | Top-1 Agree | Final Loss |
|---|---|---|---|---|---|---|
| Original Model | All | $2.191 \pm 0.058$ | $0.276 \pm 0.015$ | $0.704 \pm 0.018$ | $1.00$ | – |
| Hyperfitting | All | $0.792 \pm 0.026$ | $\mathbf{0.610} \pm 0.019$ | $0.244 \pm 0.017$ | $0.629 \pm 0.020$ | $\approx 0.00$ |
| Full LoRA | All (0–26) | $0.909 \pm 0.028$ | $0.559 \pm 0.019$ | $\mathbf{0.241} \pm 0.017$ | $0.523 \pm 0.022$ | $0.037$ |
| Late-Stage LoRA | Last 5 (22–26) | $0.893 \pm 0.027$ | $0.608 \pm 0.019$ | $0.242 \pm 0.017$ | $\mathbf{0.522} \pm 0.022$ | $0.040$ |

**Optimization Equivalence:** Figure 12 shows that Late-Stage LoRA (updating only Layers 22–26) converges with the same velocity and final loss ($\mathcal{L} \approx 0.040$) as Full LoRA.

**Terminal Sufficiency:** Table 14 reveals a remarkable finding. The Late-Stage LoRA model achieves a TTR of 0.608, which substantially outperforms the Full LoRA baseline (0.559) and essentially matches full-parameter Hyperfitting (0.610). By avoiding updates to the early accumulation layers, the Late-Stage approach appears to isolate the diversity mechanism more cleanly.

**Conclusion:** This result reinforces our "Terminal Sufficiency" hypothesis: the capacity to solve greedy degeneration is entirely contained within the final effective-dimensional expansion. Updating the preceding 22 layers is not only computationally wasteful but, in the case of Gemma, potentially suboptimal compared to a focused terminal intervention.

# E. LLaMA-3.1-8B-Instruct Model Results

Based on the summaries and mechanistic insights established in the preceding sections (Appendices B, C, and D), we continued our investigation by applying the Late-Stage LoRA strategy to significantly larger-scale architectures. Specifically, we examine the LLaMA-3.1-8B-Instruct (AI@Meta, 2024) model to verify whether the "Terminal Sufficiency" hypothesis holds when the model parameter count increases nearly fourfold and, crucially, when the base model has already undergone Supervised Fine-Tuning (SFT).

### E.1. Optimization Dynamics: Synchronized Volatility

We first analyze the training stability and convergence behavior in the 8B parameter regime. Figure 13 illustrates the loss landscapes of the standard Full LoRA (updating all identified layers 0–32) versus our targeted Late-Stage LoRA (updating only the "active zone," layers 28–32).

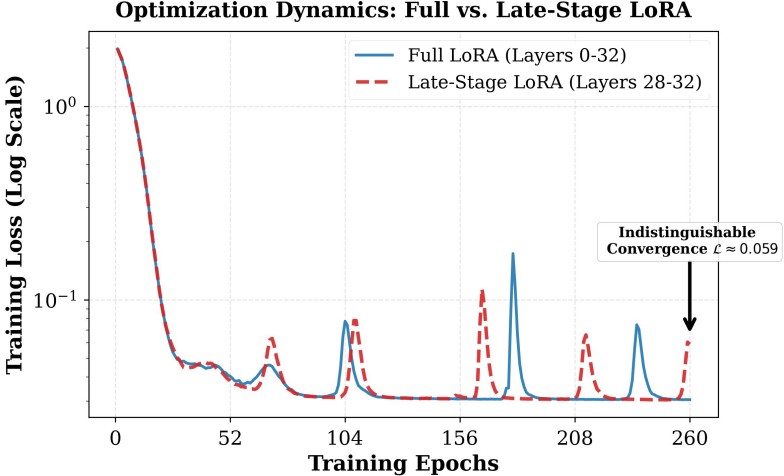

*Figure 13.* **Convergence Efficiency.** Training loss trajectories for Full LoRA (Blue) vs. Late-Stage LoRA (Red Dashed). Despite freezing the first 28 layers, the Late-Stage model converges to the same low-loss regime ($\mathcal{L} \approx 0.059$) as the Full LoRA model, indicating that the optimization capacity required for hyperfitting is fully contained within the terminal layers.

**Analysis of Optimization Isomorphism:** Unlike the smooth convergence observed in smaller base models, the 8B-Instruct model exhibits distinct *Optimization Volatility*, characterized by periodic loss spikes (e.g., at epochs $\sim 104$, $\sim 170$, and $\sim 260$). This behavior is typical when fine-tuning aligned models, representing the tension between the model's robust pre-trained priors and the Hyperfitting objective.

**Notably,** Figure 13 reveals a remarkable topological synchronization. The Late-Stage LoRA trajectory (Red Dashed) mirrors the Full LoRA trajectory (Blue Solid) with high fidelity. Even during periods of drastic gradient variance (loss spikes), the Late-Stage model reacts indistinguishably from the Full LoRA baseline. This "Iso-dynamical" behavior provides empirical evidence that the early layers of the network act as a rigid structural backbone. The capacity to navigate these complex dynamics and recover from shocks is fully encapsulated in the terminal layers. Both models ultimately converge to an indistinguishable loss floor ($\mathcal{L} \approx 0.059$), validating that the terminal intervention is robust even in volatile optimization landscapes.

### E.2. The Stability-Plasticity Trade-off

Table 15 presents the quantitative performance metrics. In the context of an Instruction-Tuned model, the results highlight a distinct advantage of the Late-Stage approach regarding Alignment Preservation.

*Table 15.* Scalability Analysis on LLaMA-3.1-8B-Instruct. Comparison of Full LoRA vs. Late-Stage LoRA. While Full LoRA achieves marginally higher diversity, it comes at the cost of significantly lower entropy (0.620). Late-Stage LoRA preserves higher entropy (0.974) and Top-1 Agreement (0.597), indicating better preservation of the base model's instructional knowledge while still effectively suppressing repetition.

| Model Variant | Layers Tuned | Entropy | TTR ↑ | Bigram Rep. ↓ | Top-1 Agree | Final Loss |
|---|---|---|---|---|---|---|
| Full LoRA | All (0–32) | $0.620 \pm 0.022$ | **0.684** $\pm 0.020$ | **0.166** $\pm 0.016$ | $0.528 \pm 0.022$ | 0.031 |
| Late-Stage LoRA | Last 5 (28–32) | $0.974 \pm 0.029$ | $0.667 \pm 0.020$ | $0.182 \pm 0.017$ | **0.597** $\pm 0.022$ | 0.033 |

**Diversity Parity:** The Late-Stage LoRA model achieves a Type-Token Ratio (TTR) of 0.667, retaining 97.5% of the

diversity capacity of the Full LoRA baseline (0.684). The Bigram Repetition (0.182) remains low, suggesting that the geometric expansion required to break degenerate loops functions effectively at the 8B scale.

**Alignment Preservation (The Conservative Advantage):** A notable divergence appears in the Top-1 Agreement. The Late-Stage model retains a significantly higher agreement with the original backbone (0.597) compared to Full LoRA (0.528). Furthermore, Full LoRA exhibits a sharp drop in prediction entropy (0.620), suggesting a tendency towards overfitting or mode collapse.

## F. Qwen2.5-7B-Instruct Model Results

Following the summary of our findings on LLaMA-3.1-8B-Instruct, we concluded our empirical investigation by applying the Late-Stage LoRA strategy to the Qwen2.5-7B-Instruct model (Qwen Team, 2024). This final experiment serves as a stress test, evaluating whether the "Terminal Sufficiency" hypothesis holds on a model that combines a larger scale with the aggressive post-training alignment characteristic of the Qwen family.

### F.1. Optimization Dynamics: The Efficiency of Isolation

We visualize the training trajectory in Figure 14. The Qwen2.5-7B-Instruct model exhibits a convergence profile distinct from smaller models, revealing a phenomenon we term "Optimization Interference" in full-parameter adaptation.

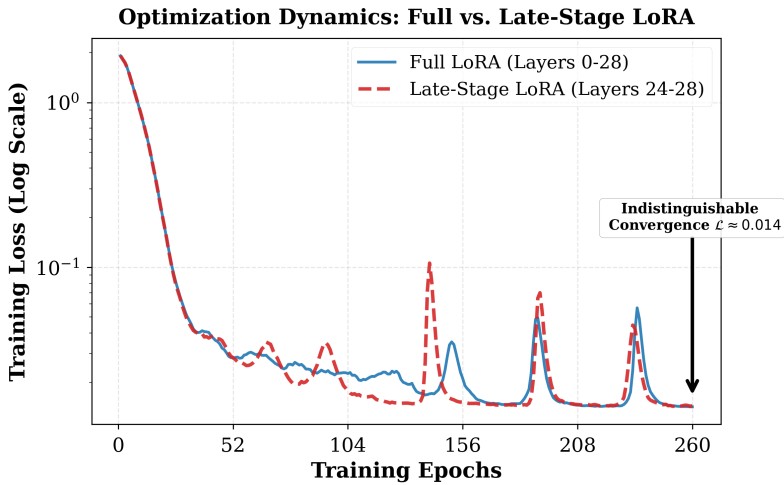

*Figure 14.* **Convergence Efficiency.** Training loss trajectories for Full LoRA (Blue) vs. Late-Stage LoRA (Red Dashed). Despite freezing the first 24 layers, the Late-Stage model converges to the same low-loss regime ($\mathcal{L} \approx 0.014$) as the Full LoRA model, indicating that the optimization capacity required for hyperfitting is fully contained within the terminal layers.

**Mitigation of Optimization Drag:** Contrary to the intuition that more trainable parameters yield faster convergence, observe the interval between Epoch 50 and Epoch 130. During this phase, the Late-Stage LoRA (Red Dashed) consistently achieves a lower training loss than the Full LoRA baseline (Blue Solid).

Despite distinct trajectories, both models eventually synchronize to an identical, exceptionally low loss floor of $\mathcal{L} \approx 0.014$. The transient spikes observed in the Late-Stage model (e.g., Epoch $\sim$135) reflect Optimization Strain, the geometric pressure of forcing the terminal layers to absorb the entire optimization load. The model's ability to recover from these spikes and match the baseline's convergence is an indicator of the robustness of the terminal intervention.

### F.2. Metrics Analysis: Prevention of Confidence Collapse

Table 16 details the generation performance. The comparison highlights a crucial safety advantage of the Late-Stage approach: mitigating *Confidence Collapse*.

**Entropy Preservation:** A notable disparity is observed in prediction entropy. The Full LoRA model exhibits a sharp drop

*Table 16.* Scalability Analysis on Qwen2.5-7B-Instruct. Comparison of Full LoRA vs. Late-Stage LoRA. Key Observation: Full LoRA suffers from a significant drop in entropy (0.733), suggesting potential overfitting or confidence collapse. Late-Stage LoRA maintains a healthier distribution (Entropy 1.219) while achieving identical convergence ($\mathcal{L} = 0.014$) and effectively suppressing repetition (Bigram Rep. 0.213).

| Model Variant | Layers Tuned | Entropy | TTR ↑ | Bigram Rep. ↓ | Top-1 Agree | Final Loss |
|---|---|---|---|---|---|---|
| Full LoRA | All (0–28) | $0.733 \pm 0.024$ | **0.672** $\pm 0.020$ | **0.192** $\pm 0.017$ | $0.528 \pm 0.022$ | 0.014 |
| Late-Stage LoRA | Last 5 (24–28) | $1.219 \pm 0.033$ | $0.592 \pm 0.019$ | $0.213 \pm 0.017$ | **0.459** $\pm 0.022$ | 0.014 |

to 0.733 nats, indicating that it has become extremely peaked and potentially overconfident in its predictions—a known precursor to mode collapse on small datasets. In contrast, Late-Stage LoRA maintains a much healthier entropy of 1.219 nats.

**Aggressive Reordering:** The Top-1 Agreement for Late-Stage LoRA is 0.459, which is notably lower than Full LoRA's 0.528. This indicates that, to compensate for the frozen backbone, the terminal layers in Late-Stage LoRA perform a more aggressive affine transformation to correct the ranking, successfully "dislodging" deeply embedded repetitive tokens from the pre-trained priors.

# G. LLM-as-Judge

**LLM-as-Judge Prompt.** We use a single-turn LLM-as-judge protocol that compares two continuations given the same context, scoring each on *coherence* and *diversity*. The judge is instructed to select a winner (A, B, or tie) and to return integer scores in $[1, 10]$ for both criteria, together with a brief rationale. To ensure reliable parsing, the judge output is constrained to a *JSON-only* response with a fixed schema:

```
You are a professional expert in text quality evaluation.  Please assess the
quality of the following two text continuations.

**Context:**
{context}

**Continuation A (Last Generated):**
{continuation_a}

**Continuation B (LoRA Generated):**
{continuation_b}

**Evaluation Criteria:**
1.  **Coherence:** Does the continuation naturally follow from the context?  Is
the logic fluent and consistent in terms of topic and style?
2.  **Diversity:** Does the continuation provide rich information, avoid excessive
repetition, and exhibit varied vocabulary and sentence structures?

**Evaluation Instructions:**
 - Carefully compare the coherence and diversity of both continuations.
 - Consider which continuation better extends the context while delivering richer
 and more meaningful content.
 - If the two continuations are of comparable quality and no clear winner can be
 determined, declare a tie.

**Please output your evaluation strictly in the following JSON format:**
{
"winner":  "A" or "B" or "tie",
"coherence_score_a":  integer score from 1 to 10,
"coherence_score_b":  integer score from 1 to 10,
"diversity_score_a":  integer score from 1 to 10,
"diversity_score_b":  integer score from 1 to 10,
"reasoning":  "your detailed rationale"
}

Output only the JSON object.  Do not include any other text.
```

