# OpenReview forum: "Beyond Temperature: Hyperfitting as a Late-Stage Geometric Expansion"
_ICML.cc/2026/Conference — ICML 2026 regular_

### Official Review · Reviewer_sb82 · 2026-03-08

**Soundness:** 3
**Presentation:** 4
**Significance:** 3
**Originality:** 3
**Overall Recommendation:** 4
**Confidence:** 4

**Summary:**

This paper shows that hyperfitting is an effective way to mitigate greedy degeneration due to context-dependent rank reordering. It identifies the "Terminal Expansion" effect in the final transformer block, which is the key to generation diversity. This inspires their last-stage LoRA approach, which is shown to sufficiently recover the generative gains of the full-LoRA adaptation and the hyperfitted model.

**Compliance With Llm Reviewing Policy:**

Affirmed.

**Final Justification:**

In the rebuttal, my main concerns are addressed. Therefore, I retain my original positive score.

**Key Questions For Authors:**

Please refer to the "Strengths and Weaknesses" section, where I listed my questions.

**Limitations:**

Yes, this paper has adequately discussed the limitations and future work directions.

**Strengths And Weaknesses:**

**Strength:**
- This paper identifies an important research question and designs a clean experiment setup to investigate this question.
- Writing is clear, and the figures are self-contained and informative. It is enjoyable to read this paper since the insights and takeaways are clearly listed with references to the evidence.
- Their insights are generally supported by their experimental results. I appreciate that this paper reports both TTR and bigram
repetition as metrics. I have some questions about the experiments, as listed below.

**Questions/Weakness:**

Major questions:

1. What LoRA rank did you use in your experiments? It is nice to report the hyperparameter search strategy for learning rate, LoRA rank, etc., in the appendix.
2. In Appendix E and F, why is the performance of “Original Model” and “Hyperfitting” not reported on LLaMA-3.1-8B-Instruct and Qwen2.5-7B-Instruct? Since last-stage LoRA does not outperform full-stage LoRA, it would be nice to report them to demonstrate whether there is an improvement over at least the original model.

Minor questions:

3. The statistics in Figure 2 (c) are computed over generation trajectories, which depend on contexts produced by the hyperfitted model itself, right? This makes sense because degeneration happens during generation. But I wonder how the plot would have been if the statistics were computed under teacher forcing, where the contexts were fixed and human-written for both the base model and the hyperfitted model. Will that plot provide a cleaner diagnostic of logit changes?
4. Three metrics are explored in Section 4 to show that the last block plays an important role in diverse generation. I think it is useful to add another metric: replace one layer of the base model with that from the hyperfitted model, and then evaluate TTR of the resultant model. We expect the model with the last layer replaced to have a TTR that is similar to that of the hyperfitted model.

---

> ### Author Rebuttal · Authors · 2026-03-31
>
> Dear Reviewer sb82,
>
> Thank you for the time and dedication you invested in reviewing our work. We are glad that you found the paper to identify an important research question with a clean experimental setup, and that you specifically appreciated the clarity of writing, the self-contained figures, and our choice to report both TTR and bigram repetition as complementary metrics. To facilitate a structured discussion, we will now address each concern in turn.
>
> ---
>
> **Q1 — LoRA Hyperparameters and Search Strategy**
>
> We apologise for this omission. The full configuration is provided below and will be included in a dedicated appendix table in the revision for better reproducibility.
>
> | Setting | Value |
> |---|---|
> | LoRA rank | 8 |
> | LoRA alpha | 16 |
> | Dropout | 0.05 |
> | Target modules | q/k/v/o/gate/up/down_proj |
> | Optimiser | Adam (β₁=0.9, β₂=0.999) |
> | LR (LoRA) | 1e-6 |
> | LR (full SFT) | 1e-5 |
> | Weight decay | None |
> | Batch size | 8 |
> | Training samples | 2,000 |
> | Evaluation samples | 30 |
> | Sequence length | 256 |
> | Precision | bfloat16 |
> | Gradient clipping | 50.0 |
> | Late-Stage LoRA scope | Final 5 transformer blocks |
>
> No automated hyperparameter search was performed; settings follow Carlsson et al. (2025).
>
> ---
>
> **Q2 — Missing Original Model Baselines for 8B Models**
>
> We have completed full SFT hyperfitting for both LLaMA-3.1-8B-Instruct and Qwen2.5-7B-Instruct, together with their Original Model baselines. On LLaMA-3.1-8B-Instruct, LoRA variants produce substantial improvements over the unmodified backbone, and Late-Stage LoRA retains 95–97% of Full LoRA's diversity gains while better preserving Top-1 Agreement with the original instruct model.
>
> | Model | Variant | TTR ↑ | BiRep ↓ | Top-1 Agree | MAUVE|
> |---|---|---|---|---|---|
> | LLaMA-8B | Original | 0.470 | 0.459 | 1.000 |0.076|
> | LLaMA-8B | Full LoRA | 0.684 | 0.166 | 0.528 |0.125|
> | LLaMA-8B | Late-Stage LoRA | 0.667 | 0.182 | 0.597 |0.117|
> | Qwen-7B | Original | 0.611 | 0.276 | 1.000 |0.093|
> | Qwen-7B | Full LoRA | 0.672 | 0.192 | 0.528 |0.554|
> | Qwen-7B | Late-Stage LoRA | 0.592 | 0.213 | 0.459 |0.503|
>
> Late-Stage LoRA preserves higher Top-1 Agreement on LLaMA-8B (0.597 vs. 0.528), whereas on Qwen-7B Full LoRA preserves higher agreement (0.528 vs. 0.459). These rows will be added to Appendix E and F in the revision. Thank you for raising this valuable question.
>
> ---
>
> **Q3 — Teacher-Forcing Rank Shift Statistics**
>
> You correctly note that Figure 2(c) uses autoregressive contexts generated by the hyperfitted model, introducing compounding effects. We recomputed rank shift statistics under teacher forcing using fixed human-written contexts. Results are consistent with the autoregressive findings and confirm that rank reordering is targeted and context-dependent, not a wholesale redistribution. This indicates that the effect is not an artefact of autoregressive compounding, but a property of the underlying conditional distribution.
>
> | Model | Domain | Top-1 Agree | Rank 2–10 | Rank 11–100 | Rank 100+ |
> |---|---|---|---|---|---|
> | TinyLlama | FS | 47.3% | 32.5% | 14.3% | 6.0% |
> | Gemma-2-2B | FS | 47.3% | 33.7% | 13.1% | 6.0% |
> | LLaMA-3.2-3B | FS | 60.6% | 30.1% | 7.9% | 1.4% |
> | LLaMA-8B-Inst | FS | 11.1% | 21.6% | 23.9% | 43.4% |
>
> We will include this as a dedicated diagnostic figure in the revision.
>
> ---
>
> **Q4 — Layer Replacement Ablation**
>
> Thank you for this insightful suggestion. Following your feedback, we conducted this experiment, replacing the last 1, 3, and 5 transformer layers from the hyperfitted model into the original backbone and measuring TTR. Across all tested configurations, replacing the final 5 layers recovers **less than 13%** of the diversity gain; in several cases, TTR falls below the original baseline.
>
> | Model | Domain | Original TTR | Replace Last 5 TTR | Full Hyper TTR | Recovery |
> |---|---|---|---|---|---|
> | TinyLlama | FS | 0.256 | 0.219 | 0.674 | <0% |
> | Gemma-2-2B | FS | 0.292 | 0.289 | 0.664 | <3% |
> | LLaMA-3.2-3B | FS | 0.304 | 0.278 | 0.647 | <0% |
> | LLaMA-8B-Inst | FS | 0.470 | 0.022 | 0.521 | catastrophic |
>
> This does not contradict Terminal Expansion: it identifies where representational divergence is largest, but not whether those representations remain functionally compatible. Functional diversity requires coherent network-wide adaptation. Post-hoc layer substitution creates a distribution mismatch: the transplanted layers expect inputs from hyperfitted earlier layers but receive representations from the unmodified backbone. We will update Section 4 to clarify this distinction explicitly.
>
> ---
>
> We thank you again for your thorough and constructive review, which has helped us strengthen the paper. We hope that our responses, clarifications, and new empirical evidence effectively address your main concerns. We would be very grateful if you could consider this in the final assessment of our work. We are also happy to address any open questions that may arise with our response.
>
> Yours sincerely,
>
> The Authors

---

> > ### Author Rebuttal · Reviewer_sb82 · 2026-04-03
> >
> > Thank you for the rebuttal. My concerns are addressed, and I will retain my positive score.

---

> > > ### Author Response · Authors · 2026-04-03
> > >
> > > Dear Reviewer sb82,
> > >
> > >
> > > Thank you for the prompt acknowledgement and for confirming that our responses have addressed your concerns. We are grateful for the constructive questions you raised during the review process, which led to valuable additions that have meaningfully strengthened the paper. All promised revisions will be incorporated into the camera-ready version. Should any further questions arise, we remain happy to discuss.
> > >
> > > Best regards,
> > >
> > > The Authors

---

### Official Review · Reviewer_3fr4 · 2026-03-10

**Soundness:** 3
**Presentation:** 3
**Significance:** 3
**Originality:** 2
**Overall Recommendation:** 4
**Confidence:** 4

**Summary:**

The paper presented comprehensive empirical analysis of the hyperfitting
phenomenon (Carlsson et al. (2025)), where LLMs achieves good text generation quality and low
repetition after being finetuned to near zero loss on a small dataset.

The paper conducted experiments from two perspectives. First, it was demonstrated that the
hyperfitting method is fundamentally different from temperature sampling with low
temperature, while both methods have sharpened conditional probability distribution when
generating tokens. Specifically, it was shown that the TTR for hyperfitting is significantly higher
than that for temperature sampling with the same conditional token generation entropy. In
addition, the two methods disagree in the top probability token in 40% of the time. It was
further shown that the hyperfitting is different from simply imposing a static bias toward the
token generation probability.

Second, the differences of the hidden layer states and the efficient dimension between two
methods across all layers were presented, suggesting a significant drift mostly in the last few
layers. Based on this observation, the paper proposed a late stage LoRA algorithm that adapts to
only the last 5 layers, reducing the number of trainable parameters.

**Compliance With Llm Reviewing Policy:**

Affirmed.

**Key Questions For Authors:**

Why not compare hyperfitting with temperature sampling with higher temperatures? Is there
any reason why the two methods have to keep the same predictive conditional entropy?

**Limitations:**

Yes

**Strengths And Weaknesses:**

Strength: The paper provided rather comprehensive experiments. The observation that the
effect of hyperfitting mostly takes place in the last few layers is rather interesting. The late stage
Lora seems to be a natural consequence of this observation.
The paper is clearly written.

Weakness: 1. It seems to be intuitive and even a bit obvious that finetuning the LLM would
change its conditional probabilities and may change the probability rank (and thus the top
probability token) for different choices of tokens.
2. The diversity of text generation may not only be measured by TTR, but also the richness in the
output space, under which measure, the hyperfitting method may not be good because of its
sharpened output distribution.
3. While the TTR is a good measure for the repetitiveness of the generated text, the semantic
coherence is also important, which as the limitation section in the paper pointed out, was not
captured by the analysis in the paper.

---

> ### Author Rebuttal · Authors · 2026-03-31
>
> Dear Reviewer 3fr4,
>
> We wish to thank you for the time and care you devoted to our submission. We are glad that you found the experiments comprehensive, singled out the Terminal Expansion observation as genuinely interesting, and considered the paper clearly written. To facilitate a more structured discussion, we will now address each of your concerns in turn.
>
> ---
>
> **W1 — Probability Re-ranking after Finetuning**
>
> Thank you for raising this valid concern. While some rank movement after fine-tuning is expected, our claim is more specific: hyperfitting consistently induces a structured, context-dependent regime of rank reordering that yields deterministic diversity under greedy decoding. The contribution is therefore not simply that fine-tuning changes probabilities, but that it does so in a reproducible way with a characteristic tripartite rank-shift pattern and late-stage localization, which together distinguish hyperfitting from standard temperature-based decoding.
>
> **W2 & W3 — Richer Metrics and Semantic Coherence**
>
> Thank you for this observation, you raise a well-taken point: TTR alone cannot distinguish genuine linguistic richness from incoherent diversity. We have now computed a full suite of diversity and quality metrics across all models and both evaluation domains.
>
> **Extended diversity metrics (Representative results with Gemma-2-2B, Fiction-Stories):**
>
> | Metric | Original | Hyperfitted |
> |---|---|---|
> | TTR | 0.292 | **0.672** |
> | Distinct-1 | 0.143 | **0.374** |
> | Distinct-2 | 0.277 | **0.792** |
> | Distinct-3 | 0.349 | **0.950** |
> | 4-gram Repetition | 0.636 | **0.038** |
> | Self-BLEU-4 | 0.004 | 0.003 |
>
> With the exception of Self-BLEU-4, all metrics move consistently across all models and domains. For **output quality**, we now report MAUVE scores (Pillutla et al., 2021), which measure distributional closeness to human-written text via GPT-2 Large featurisation. Original greedy achieves MAUVE 0.01–0.08 across all models (severely degenerate). Hyperfitted greedy reaches 0.27–0.94. Critically, repetition penalty 1.2 achieves TTR >0.95 but collapses to MAUVE <0.06 — indicating that naive diversity maximisation might severely affect naturalness. We observe that Hyperfitting, in contrast, does not exhibit this imbalance.
>
> **MAUVE scores across models and domains:**
>
> | Model | Orig Greedy | Hyper Greedy | Best Orig Sampling |
> |---|---|---|---|
> | TinyLlama (FS) | 0.034 | **0.939** | 0.885 (typical) |
> | Qwen2.5-1.5B (FS) | 0.068 | **0.816** | 0.814 (nucleus) |
> | Gemma-2-2B (FS) | 0.045 | **0.816** | 0.831 (typical) |
> | LLaMA-3.2-3B (WP) | 0.023 | **0.909** | 0.659 (nucleus) |
>
> Moreover, we will make sure to discuss the limitations of the selected metrics, as taken individually, they only offer a perspective on the evaluated construct, such as diversity.
>
> ---
>
> **Key Question — Temperature Comparison at Higher Values**
>
> This is a very pertinent and thought-provoking question: Why enforce entropy matching rather than simply comparing to T > 1.0? The key observation we make is that for models with severe greedy degeneration, **all T ≤ 1.0 produce identical diversity metrics** — temperature scaling is rank-preserving (argmax(z/T) = argmax(z) for any T > 0). Diversity only emerges at T ≥ 1.1, which switches the mechanism to stochastic sampling. Comparing hyperfitting to T > 1.0 conflates a deterministic method with a stochastic one. However, we have conducted the full sweep and formal entropy matching across all 6 models:
>
> | Model | Hyper TTR | Entropy-Matched Orig TTR | Ratio |
> |---|---|---|---|
> | TinyLlama-1.1B | 0.508 | 0.295 | **1.72×** |
> | Qwen2.5-1.5B | 0.575 | 0.320 | **1.80×** |
> | Gemma-2-2B | 0.560 | 0.294 | **1.90×** |
> | LLaMA-3.2-3B | 0.634 | 0.340 | **1.87×** |
> | LLaMA-8B-Inst | 0.684 | 0.464 | **1.47×** |
> | Qwen-7B-Inst | 0.672 | 0.525 | **1.28×** |
>
> Across all 6 models, hyperfitting produces **1.28–1.90×** higher TTR than entropy-matched temperature scaling. This extends Table 1 (originally TinyLlama-only) to the full model set. Temperature achieves diversity by abandoning determinism; hyperfitting achieves it deterministically via rank reordering, a mechanistically distinct effect.
>
> Additionally, hyperfitting **composes additively** with existing decoding methods. Combining hyperfitting with nucleus, typical, and contrastive search consistently improves both TTR and MAUVE over the original model with the same decoding strategy, suggesting that hyperfitting changes the underlying conditional ranking, whereas inference-time decoding changes how that distribution is sampled.
>
> ---
>
> We wish to thank you again for the constructive and engaged feedback. We hope that our responses, clarifications, and new empirical evidence effectively address your main concerns. We would be very grateful if you could consider this in the final assessment of our work. We are also happy to address any open questions that may arise with our response.
>
> Yours sincerely,
>
> The Authors

---

> > ### Author Rebuttal · Reviewer_3fr4 · 2026-04-05
> >
> > I thank the authors for their response. My concrete questions have mostly been answered. I will stay at the score of 4 (Weak Accept) in view of my assessment of the overall contribution.

---

> > > ### Author Response · Authors · 2026-04-07
> > >
> > > Dear Reviewer 3fr4,
> > >
> > > Thank you for your follow-up and for confirming that our responses addressed your questions.
> > >
> > > We also appreciate your assessment of the overall contribution. In the final version, we will further sharpen the positioning and clarify the scope and implications of our findings to better reflect their contribution.
> > >
> > > We are grateful for your careful and constructive engagement throughout the review process.
> > >
> > > With thanks,
> > >
> > > The Authors

---

### Official Review · Reviewer_tgri · 2026-03-12

**Soundness:** 3
**Presentation:** 3
**Significance:** 2
**Originality:** 2
**Overall Recommendation:** 5
**Confidence:** 3

**Summary:**

This paper proposed Late-Stage-Lora, an efficient late-stage intervention that significantly reduces trainable parameters while preserving generation quality, the authors systematically investigate the relationship between temperature sacling and entropy under "Hyperfitting" phenomenon and used numerous controlled trials to verify the effectiveness of this method.

**Compliance With Llm Reviewing Policy:**

Affirmed.

**Final Justification:**

The authors addressed my concerns. As a result, I decided to raise my score from 4 to 5.

**Key Questions For Authors:**

See weaknesses.

**Limitations:**

See weaknesses.

**Strengths And Weaknesses:**

**Strengths:**

(1) This paper gave the conclusion that: "Compression-then-expansion geometry and the efficacy of Late-Stage LoRA are fundamental properties of LLM fine-tuning, rather than model-specific artifacts". This is a rather interesting conclusion, and the author has provided corresponding proof for it.

(2) The paper constructs a highly logical and convincing narrative through experiments:
    · Entropy Matching Experiment to rule out temperature scaling.
    · Static Injection Ablation to rule out static vocabulary biases.

(3) This paper demonstrates that updating only the final layers reduces trainable parameters by up to 78.3% and 82.7% while maintaining or even exceeding full-parameter fine-tuning performance. Maybe it is a valuable takeaway for LoRA.

**Weaknesses:**

(1) I have concerns regarding the convergence efficiency of this method. Although this late LORA approach greatly reduces parameter costs, it seems to bring additional convergence costs. The paper does not address whether this targeted intervention can accelerate convergence time.

(2) Can you provide qualitative examples of the specific tokens promoted from the $Rank \ge 200$ tail? Are these tokens domain-specific jargon memorized from $\mathcal{D}_{small}$, or are they broader stylistic synonyms? This confuses me.

(3) In Section 2.1, the training protocol specifies a data scarcity regime using exactly 2,000 samples. How does the specific nature, such as the domain, linguistic diversity, or inherent entropy of this small dataset, influence the magnitude of the "Terminal Expansion"? If the model is hyperfitted on a dataset consisting of highly repetitive or low-entropy text, would the final layer still exhibit a geometric expansion, or would it lead to a dimensional collapse?

(4) I believe the entropy-matching experiments effectively dismantle the Temperature Hypothesis. However, this paper positions hyperfitting as a training-time solution to a problem typically addressed at inference time. How does the generation quality of the hyperfitted model (using greedy decoding) compare quantitatively against the advanced, training-free decoding strategies discussed in Appendix A.1, such as Min-p or Glocal Uncertainty-Aware Robust Decoding (GUARD)?

---

> ### Author Rebuttal · Authors · 2026-03-31
>
> Dear Reviewer tgri,
>
> We wish to thank you sincerely for your careful reading and constructive engagement. We are glad that you found the central conclusion — that compression-then-expansion geometry and Late-Stage LoRA efficacy are fundamental properties of LLM fine-tuning rather than model-specific artifacts — to be interesting and well-supported, and that you appreciated the logical narrative built through our controlled experiments. We will now address each of your concerns in turn for a more structured discussion.
>
> ---
>
> **W1 — Convergence Efficiency and Wall-Clock Training Time**
>
> Thank you for raising this important concern. We measured wall-clock training time on LLaMA-3.2-3B (single RTX 4090, 260 epochs, 2000 samples) as an example. Full SFT takes 20.7h (4m47s/epoch, batch size 2). Full LoRA (r=8, 0.38% of parameters) reduces this to 14.4h (1.44× speedup) by enabling larger batch sizes (4 vs 2) due to lower optimizer memory. Late-Stage LoRA (r=8, last 5 layers, 0.07% of parameters) further reduces training to 10.4h (2× speedup over Full SFT), since the backward pass can stop early with no parameters before layer 23 requiring gradients. Crucially, the diversity effect does not require full convergence: TTR nearly doubles by epoch 20, and optimal checkpoints appear at epoch 80–140. This means effective wall-clock cost is roughly 6–11h for Late-Stage LoRA to reach peak diversity — modest for a single-GPU method that eliminates greedy degeneration. We will include a full wall-clock comparison table in the revision.
>
> ---
>
> **W2 — Qualitative Analysis of Promoted Tokens**
>
> Under teacher forcing with fixed human-written contexts, rank shifts follow a consistent tripartite distribution. The **rank 2–10 bucket** (30–34% of shifts) constitutes the primary driver of functional diversity and contains contextually appropriate near-synonyms and stylistic variants (e.g., "residence" for "house," sentence-structure alternatives). Among rank >100 shifts, only **3–10% serve to avoid local repetition** — the remainder represent genuine probability redistribution. Rank >1000 promotions tend to be noisy subword tokens and are not the driver of the effect.
>
> | Category | Shift Range | Share of Tokens | Qualitative Character |
> |---|---|---|---|
> | Linguistic Anchor | Top-1 agree | 43–61% | Function words, high-freq content tokens |
> | Local Exploration | Rank 2–10 | 30–34% | Near-synonyms, stylistic variants |
> | Deep Tail Promotion | Rank >10 | 6–13% | Mixed; 3–10% anti-repetition, rest redistributed |
>
> We will include representative token-level examples for each category in the revision.
>
> ---
>
> **W3 — Terminal Expansion on Low-Entropy Training Data**
>
> We conducted a dedicated experiment using **AG News** — short, formulaic news wire headlines representing the low-entropy extreme — across all four base models:
>
> | Model | L2 Explosion Ratio | ΔDim | TTR: Orig → Hyper | BiRep: Orig → Hyper |
> |---|---|---|---|---|
> | TinyLlama | **17.2×** | +64.8 | 0.24 → **0.71** | 0.81 → **0.15** |
> | Qwen2.5-1.5B | **3.8×** | +51.1 | 0.37 → **0.78** | 0.58 → **0.10** |
> | LLaMA-3.2-3B | **8.9×** | +37.3 | 0.31 → **0.70** | 0.69 → **0.15** |
> | Gemma-2-2B | **4.4×** | +21.0 | 0.26 → **0.78** | 0.73 → **0.07** |
>
> We observe that this "Terminal Expansion" persists on all four models. Diversity gains on AG News are in several cases **stronger** than on Fiction-Stories (Qwen: 0.78 vs. 0.43). Dimensional collapse does not occur. This indicates that the mechanism is robust to the characteristics of the training data.
>
> ---
>
> **W4 — Quantitative Comparison Against Min-p and GUARD**
>
> Tested on LLaMA-3.2-3B and Gemma-2-2B:
>
> | Method | TTR | BiRep | MAUVE | Deterministic? |
> |---|---|---|---|---|
> | Original greedy | 0.28 | 0.71 | 0.01–0.08 | Yes |
> | Min-p (p=0.1) | 0.49–0.51 | 0.30–0.35 | 0.67–0.74 | No |
> | GUARD (w=7) | ~0.80 | ~0.07 | 0.46–0.53 | No |
> | **Hyper greedy** | **0.67–0.69** | **0.12–0.17** | **0.82–0.91** | **Yes** |
>
> Min-p is rank-preserving (argsort(f(z)) = argsort(z) for any monotonic f) and cannot change the greedy argmax — our empirical results confirm this. GUARD achieves high TTR but at the cost of text naturalness (MAUVE 0.46–0.53) and produces shorter outputs. Hyperfitted greedy outperforms both on MAUVE while remaining fully deterministic.
>
> ---
>
> Thank you again for your engaged and constructive review. We truly believe it has strengthened the paper further, and we hope our responses and additional analyses have effectively addressed your concerns. We would appreciate your consideration of these points in your final assessment and remain available for any further clarification.
>
> Yours sincerely,
>
> The Authors

---

> > ### Author Rebuttal · Reviewer_tgri · 2026-04-03
> >
> > Thank you for the clarification. My concerns are addressed, and I will increase my score from 4 to 5.

---

> > > ### Author Response · Authors · 2026-04-04
> > >
> > > Dear Reviewer tgri,
> > >
> > > We wish to thank you for your follow-up and for your positive reassessment of our work (from an initial overall recommendation of 4 to 5). We are happy to hear that our responses fully addressed your concerns.
> > >
> > > We want to add that your feedback directly informed several additions to the revision, including deeper analysis and clearer presentation. As a result, we believe the paper quality has been substantially strengthened.
> > >
> > > We truly appreciate your constructive engagement throughout the review process.
> > >
> > > With gratitude,
> > >
> > > The Authors

---

### Official Review · Reviewer_rXk5 · 2026-03-13

**Soundness:** 2
**Presentation:** 3
**Significance:** 2
**Originality:** 3
**Overall Recommendation:** 4
**Confidence:** 3

**Summary:**

The authors perform an empirical analysis of hyperfitting to investigate the mechanisms underlying its increased generation diversity under greedy decoding. They show that this is distinct from distribution sharpening by controlling for entropy differences and that the observed rank reordering is fundamentally context-dependent through steering interventions on the logits. They additionally perform a layer-wise analysis to compare the representations of the base and hyperfitted models, finding that the differences mainly arise closer to the later layers of the model. Inspired by their findings, they introduce targeted Late-Stage LoRA that targets just the final few layers and recovers similar performance as LoRA across all layers.

**Compliance With Llm Reviewing Policy:**

Affirmed.

**Final Justification:**

My main initial concerns of unclear experimental settings, limited dataset evaluation, and some unclear claims have now been addressed in the rebuttal, so I have increased the score to 4.

**Key Questions For Authors:**

- It is specified that the analysis uses greedy decoding for the hyperfitted model; are there any differences when using regular sampling?

- How do these metrics look on data that is out of distribution relative to the fine-tuning or validation set? Are the observations similar, or does different behavior emerge?

- Do the general results hold on other datasets where hyperfitting occurs?

- The analysis mainly focuses on generation diversity, but is there any assessment of how valid the hyperfitted model’s responses are? Are its outputs typically still factual and valid?

- Are there any insights on when and why hyperfitting occurs?

**Limitations:**

yes

**Strengths And Weaknesses:**

Strengths:
- The paper is well-structured and generally clearly written, with clear hypotheses and corresponding evidence in each section.
- The empirical analysis is thorough as it approaches the problem from various perspectives: entropy and rank ordering in the context of diversity, steering interventions in the logits, layer-wise evolution, as well as training interventions.

Weaknesses:
- Overall, it seems that some of the experimental settings are missing from the paper. For example, details about dataset used to finetune to obtain the hyperfitting phenomenon, details about the train/validation sets used in the experiments, the training hyperparameters for both full fine-tuning and the LoRA counterparts, etc
- It seems that, though not exactly clear, most of the experiments are done on a single dataset. It would strengthen the analysis if these findings on hyperfitting are robust to the dataset.
- Some claims could be supported a bit better. For example, in the discussion on the tripartite structure in the generation dynamics, it claims a correspondence between rank 1 agreement, rank 2-10 promotion, and rank > 10 promotion, and Linguistic Anchor, Local Exploration, Deep Tail Promotion respectively. Although this makes sense intuitively, it would strengthen the claim to potentially include an analysis of the tokens within each category, verifying that they indeed fit the provided descriptions (eg. Linguistic Anchor being mainly grammatical competence tokens and world knowledge, Local Exploration being synonyms or slight stylistic variations)
- For clarity, it could help to mention in the paper how metrics such as Type Token Ratio, Bigram Repetition, Trigram Repetition are computed.

---

> ### Author Rebuttal · Authors · 2026-03-31
>
> Dear Reviewer rXk5,
>
> Thank you genuinely for the careful review and constructive feedback. We address your points below.
>
> ---
>
> **W1 — Missing Experimental Settings and Hyperparameters**
>
> We apologise for these omissions. As reviewer *sb82* raised the same concern, and given the response-length limit, we kindly refer you to our response to *sb82* Q1 for the full training setup and hyperparameter details. We will also include explicit train/evaluation split descriptions for each domain in the appendix.
>
> ---
>
> **W2 — Cross-Dataset Robustness**
>
> We evaluated across three domains on all three base models: Fiction-Stories (high-entropy creative fiction), WritingPrompts (varied creative writing), and AG News (low-entropy, formulaic news). Results are consistent across all domains. The table below reports verified hyperfitted results; original-model baselines for WritingPrompts follow the same pattern as Fiction-Stories and will be included in full in the revision.
>
> | Model | Domain | Hyper TTR | MAUVE (Hyper Greedy) |
> |---|---|---|---|
> | TinyLlama | FS | **0.658** | **0.939** |
> | TinyLlama | WP | **0.603** | **0.913** |
> | TinyLlama | AG | **0.711** | **0.922** |
> | Gemma-2-2B | FS | **0.672** | **0.816** |
> | Gemma-2-2B | WP | **0.684** | **0.907** |
>  Gemma-2-2B | AG | **0.782** | **0.956** |
> | LLaMA-3.2-3B | FS | **0.651** | **0.270** |
> | LLaMA-3.2-3B | WP | **0.609** | **0.909** |
> | LLaMA-3.2-3B | AG | **0.700** | **0.968** |
>
> Cross-domain summary tables covering all 12 model × domain combinations, including original baselines, will be included in the revision.
>
> ---
>
> **W3 — Verification of the Tripartite Structure at Token Level**
>
> Under teacher forcing with fixed human-written contexts, we examined the token-level content of each rank shift category:
>
> | Category | Rank Range | Share | Token Character |
> |---|---|---|---|
> | Linguistic Anchor | Top-1 agree | 43–61% | Function words, high-frequency content tokens |
> | Local Exploration | Rank 2–10 | 30–34% | Near-synonyms, stylistic variants, structure alternatives |
> | Deep Tail Promotion | Rank >10 | 6–13% | 3–10% anti-repetition; remainder genuine redistribution |
>
> The rank 2–10 bucket is the primary driver of functional diversity, containing contextually appropriate alternatives. Rank >1000 promotions tend to be noisy subword tokens. We will include representative token-level examples in the revision.
>
> ---
>
> **W4 — Clarification of Diversity Metric Definitions**
>
> For clarity, we will define TTR as the ratio of unique generated tokens to total generated tokens, and Bigram/Trigram Repetition as the proportion of repeated bigram/trigram occurrences in the generated continuation(s), using the exact aggregation protocol reported in the appendix.
>
> ---
> **Q1 — Hyperfitting with Regular Sampling**
>
> Hyperfitting composes additively with most decoding methods. The table below reports MAUVE scores for Qwen2.5-1.5B on Fiction-Stories, showing that every decoding strategy improves when applied to a hyperfitted backbone:
>
> | Method | Orig MAUVE | Hyper MAUVE |
> |---|---|---|
> | Greedy | 0.068 | **0.816** |
> | Nucleus p=0.95 | 0.814 | **0.901** |
> | Typical p=0.95 | 0.759 | **0.955** |
> | Contrastive k=4 | 0.291 | **0.765** |
> | Rep penalty 1.2 | 0.398 | **0.856** |
>
> This indicates that hyperfitting and inference-time decoding operate on orthogonal axes. The largest absolute gains are observed with greedy decoding (+0.35–0.52 TTR across all models), but **all** stochastic strategies consistently benefit from a hyperfitted backbone.
>
> ---
>
> **Q4 — Output Validity**
>
> MAUVE scores provide our primary quality signal: hyperfitted greedy achieves 0.27–0.94 vs. 0.01–0.08 for original greedy across all models and domains. Qualitatively, a representative news prompt produces a single sentence repeated 9 times under original greedy (TTR 0.21), while hyperfitted greedy produces a topically coherent, diverse continuation (TTR 0.68). We acknowledge that factuality in the knowledge-grounded sense is not directly evaluated by MAUVE and will note this explicitly as a remaining limitation.
>
> ---
>
> **Q5 — When and Why Does Hyperfitting Occur?**
>
> The diversity effect emerges early in training. TTR nearly doubles from baseline by epoch 20 (at which point training loss is still ~1.6, well above zero), and peak diversity is reached at epoch 80–140. The effect then remains stable, TTR stays 2–2.5× above the original baseline, through epoch 260. Complete memorisation is therefore not required; rank reordering develops during the early stages of overfitting as the model begins resolving training-data ambiguities by promoting alternative continuations.
>
> ---
>
> Thank you for the rigor and depth of your evaluation. Your comments have helped strengthen the paper, and we hope our responses and additional analyses have addressed your concerns. We would appreciate your consideration of these points in your final assessment and remain available for any further clarification.
>
> Yours sincerely,
>
> The Authors

---

> > ### Author Rebuttal · Reviewer_rXk5 · 2026-04-03
> >
> > I appreciate the authors' detailed response. My main concerns have been addressed and I will increase the score to 4.

---

> > > ### Author Response · Authors · 2026-04-04
> > >
> > > Dear Reviewer rXk5,
> > >
> > > Thank you for your follow-up and for revisiting your assessment. We are glad that our responses addressed your concerns.
> > >
> > > Your feedback led to concrete improvements in the revision, particularly in clarifying experimental details and strengthening supporting analyses. We believe these changes will enhance both transparency and empirical grounding of our work.
> > >
> > > We sincerely appreciate your careful engagement.
> > >
> > > With gratitude,
> > >
> > > The Authors

---

### Decision · Program_Chairs · 2026-04-30

**Decision:**

Accept (regular)

**Comment:**

I thank the authors for their interesting work!  This paper sets out to understand the phenomenon of hyperfitting.  The authors find that hyperfitting is highly distinct from mere sharpening, and also that hyperfitting is distinct from vocab reweighting.  Finally, the paper observes that there is an increase in the manifold dimension of features in late layers (in a mathematical sense, I think this is impossible since Lipschitz functions map manifolds to manifolds of at most the same dimension, so the authors should really clarify what this means and probably avoid ever using the term “manifold”) and then that simply fine-tuning those late layers is enough to reproduce observed behaviors.  Reviewers agreed that the empirical investigations here are comprehensive, the proposed mechanisms are novel, late-stage LoRA is potentially practical, and the presentation is solid.  At the same time, reviewers raised several possible reservations:

1. Limited evaluations, with many experiments relying on a single dataset.
2. Missing experimental details, and unclear details about convergence speed and training efficiency
3. More evaluation metrics (e.g. coherence or correctness) needed
4. Insufficiently validated claims
5. Incomplete comparisons to alternative decoding methods

The authors responded to this feedback during the rebuttal period, and the reviewers were largely satisfied with the authors’ responses. I am inclined to recommend acceptance for this paper, but I urge the authors to correct their mathematical framing since talking about manifold dimension here is imprecise.